# Pooled prevalence and factors of overweight/obesity among women of reproductive age in low and middle-income countries with high maternal mortality: A multi-level analysis of recent demographic and health surveys

Demiss Mulatu Geberu[1]*, Kaleab Mesfin Abera[2], Yawkal Tsega[3], Abel Endawkie[4], Wubshet D. Negash[1,5], Amare Mesfin Workie[6], Lamrot Yohannes[7], Mihret Getnet[8,9], Nigusu Worku[1], Adina Yeshambel Belay[1], Lakew Asmare[9], Hiwot Tadesse Alemu[1], Misganaw Guadie Tiruneh[1], Asebe Hagos[1], Melak Jejaw[1], Kaleb Assegid Demissie[1]

1 Department of Health Systems and Policy, Institute of Public Health, College of Medicine and Health Sciences, University of Gondar, Gondar, Ethiopia, 2 Department of Health Systems and Policy, Institute of Public Health, College of Medicine and Health Sciences, Wollo University, Dessie, Ethiopia, 3 Department of Health System and Management, School of Public Health, College of Medicine and Health Sciences, Wollo University, Dessie, Ethiopia, 4 Department of Epidemiology and Biostatistics, School of Public Health, College of Medicine and Health Sciences, Wollo University, Dessie, Ethiopia, 5 National Centre for Epidemiology and Population Health, The Australian National University, Canberra, Australia, 6 Department of Nutrition, Institute of Public Health, College of Medicine and Health Sciences, University of Gondar, Gondar, Ethiopia, 7 Department of Environmental and Occupational Health and Safety, Institute of Public Health, University of Gondar, Gondar, Ethiopia, 8 Department of Human Physiology, School of Medicine, College of Medicine and Health Sciences, University of Gondar, Gondar, Ethiopia, 9 Department of Epidemiology and Biostatistics, Institute of Public Health, College of Medicine and Health Sciences, University of Gondar, Gondar, Ethiopia

* mulatu.demissma23@gmail.com

## Abstract

### Background

Due to the increased magnitude of overweight/obesity in many countries, the World Health Organization (WHO) has identified it as a significant public health crisis, particularly affecting women of reproductive age in developing nations. Despite obesity/overweight among women of reproductive age being widely acknowledged as a pressing public health issue, there has been limited investigation into its pooled prevalence and various associated factors in low and middle-income countries (LMICs) with high maternal mortality. Thus, the objective of our study was to assess the pooled prevalence and associated factors of overweight/obesity among reproductive-age women in low and middle-income countries with high maternal mortality.

### Methods

We analyzed secondary data using recent Demographic and Health Survey datasets from 21 low and middle-income countries with high maternal mortality. A weighted sample

**Data availability statement:** The data used for the analysis of this study is attached as supplementary file 1.

**Funding:** The author(s) received no specific funding for this work.

**Competing interests:** The authors have declared that no competing interests exist.

**Abbreviations:** AOR, Adjusted Odds Ratio; BMI, body mass index; CI, Confidence Interval; CDC, Centers for Disease Control and Prevention; CHF, Congestive Heart Failure; CKD, Chronic Kidney Disease; DHSs, Demographic, and Health Surveys; DALYs, disability-adjusted life years; ICC, Intracluster correlation coefficient; IR, individual record; IRB, Institutional Review Board; LMIC, Low and Middle-Income Countries; MOR, median odds ratio; MUAC, Mid-Upper Arm Circumference; PCV, a proportional change in variance; UNICEF, United Nations International Children's Emergency Fund; VIF, Variance Inflation Factor; WHO, World Health Organization.

of 64,076 women of reproductive age was included in the analysis. The variables were extracted from the IR file, and the data were cleaned, recoded, and analyzed using STATA version 14.2 software. A multilevel binary logistic regression model was applied, and adjusted odds ratios (AOR) with 95% confidence intervals and a p-value of ≤ 0.05 were used to identify statistically significant associated factors. Model fitness and comparison were assessed using the ICC, MOR, PCV, and deviance (−2LLR).

## Result

In this study, the pooled prevalence of overweight/obesity among women of reproductive age was 32% (95% CI: 27% − 37%), with a significant variation between countries, ranging from 10% in Burundi to 53% in Mauritania. Women of reproductive age with overweight/obesity showed a significant positive association with various factors compared to those with a normal BMI. Accordingly, women's age, women's educational status, women's occupation, women's marital status, households' income levels, number of living children, frequency of watching television, using the internet, sex of household head, and sources of drinking water were identified as individual-level factors. On the other hand, residence, community poverty, and community-level media usage were found to be significantly associated with community-level variables.

## Conclusions and recommendations

More than three out of ten women of reproductive age were overweight/obese in low and middle-income countries with high maternal mortality. Individual-level and community-level factors were associated with overweight/obesity. Special attention is recommended to older women, those with formal education, non-working women, individuals who spend time watching television and using the internet, urban residents, and female household heads. Furthermore, since higher household income is associated with an increased likelihood of weight gain, it is important to provide appropriate health interventions for women from the wealthiest households.

## Background

The global prevalence of overweight and obesity is a growing public health concern, characterized by the WHO as a pandemic due to its impact in both developed and developing countries [1]. It ranks as the sixth-highest cause of disability-adjusted life years (DALYs) and leads to approximately 4.0 million annual deaths worldwide [2–4]. Additionally, there has been a 28.3% increase in global mortality related to overweight/obesity from 1990 to 2015 [3]. Studies have indicated that marked dietary changes have resulted in a faster increase in the problem in LMICs compared to higher-income countries [4,5]. Out of the two billion overweight/obese individuals globally, 62% were identified in LMICs [6].

Multiple studies have indicated a significantly higher prevalence of the condition in women as compared to men [3–5,7–11]. Despite LMICs being affected by both

overweight/obesity and underweight issues, the prevalence of overweight surpasses that of underweight among adult women [12]. A trend analysis of women of reproductive age in 39 LMICs revealed a rising prevalence of overweight/obesity over time, with the increase being more pronounced among the lowest economic groups compared to the highest in these 39 studied LMICs [13].

Several studies have shown high rates of overweight and obesity among women of reproductive age in various countries. The prevalence was reported at 55.2% in Brazil [14], 43% in Bangladesh [15], 18% in Cambodia [16], 19.4% in Timor-Leste [17], 35.5% in Ghana [18], 50.4% in Tanzania [19], and 26.9% in Mali [20]. Despite interventions such as restrictions on unhealthy food advertising, improvements in school meals, taxation to reduce consumption of unhealthy foods, and supply chain incentives to promote healthy foods, the prevalence of overweight/obesity in women of reproductive age continues to rise unacceptably [3]. Trend analyses have shown substantial increases over time, with rates in Bangladesh increasing from 9.35% in 1999 to 39.14% in 2014 [7], in Zimbabwe from 25% in 2005 to 36.6% in 2015 [21], and in Nepal from 20.5% in 2006 to 41.1% in 2016 [22].

Previous studies found that overweight/obesity directly contributed to 53% of all female deaths, serving as a well-documented risk factor for obstetric complications in both women of reproductive age and their future offspring [23]. In women of reproductive age, complications can include diabetes mellitus [8,24–27], hypertension [26–28], Chronic Kidney Disease (CKD) [3,29], Congestive Heart Failure (CHF) [3,30], cesarean delivery and surgical site infection [26,27,31–33], miscarriage and stillbirth [34], fetal macrosomia [35,36], breast and cervical cancers [27], postpartum endometritis [37], prolonged hospital stay, and postpartum hemorrhage [26,31]. For newborns, complications can include low birth weight, congenital malformations, preterm birth, large-for-gestational-age babies, and perinatal death are some of the complications [27,38,39].

Various studies have identified multiple socioeconomic and behavioral factors contributing to overweight and obesity among women of reproductive age. These factors include high socioeconomic status [1,8,10,15,18,20–22,27,40–45], age [1,9,10,14,15,19–22,27,40,41,43], parity [14,22], marital status [8,10,14,16,19–21], education [10,16,18,20,22,40,41,46,47], occupation status [46], urban residence [1,8,10,16,21,41,45], number of household members [7], contraceptive use [40,48], number of children [16,45], and frequency of watching television [17,20]. Additionally, behavioral factors such as alcohol consumption, smoking, and dietary habits play a significant role [49,50].

The rise in overweight/obesity can be attributed to the nutritional transition from traditional whole-food-based meals composed of foods like grains that have low animal source foods, salt, refined oils, and sugars [51] to energy-dense and nutrient-poor diets composed of refined carbohydrates, high-fat intake, and processed foods [52]. Studies have shown that frequent consumption of sweets, meat, and eggs, as well as snacking, contributes to increased overweight/obesity in women of reproductive age [1,8,21]. Moreover, lower levels of physical activity and certain ethnicities have been identified as determinants of overweight/obesity [8,19,27,47].

It is widely acknowledged that obesity and overweight among women of reproductive age pose a significant public health concern. However, there has been limited investigation into the pooled prevalence and contributing factors in low and middle-income countries having high maternal mortality using recent DHS data. Especially, conducting multilevel analysis is particularly valuable for deepening the understanding of complex, layered phenomena by revealing how individual- and community-level factors interact. This approach enhances both the methodological rigor and policy relevance of research in the existing literature. Additionally, multilevel analysis provides improved estimation precision and an enhanced understanding of the underlying factors. On the other hand, timely and comprehensive information on the rapid progression and health impacts of obesity and overweight is crucial for the development of effective prevention policies. Moreover, to address the mortality and morbidity associated with overweight/obesity in women of reproductive age, evidence-based interventions should be identified and implemented by healthcare providers and policymakers. Moreover, educating women and communities about potential risk factors and promoting healthy behaviors is also essential. Therefore, our study aimed to provide reliable evidence to support the WHO's goal of halting the rise in obesity by 2025 by assessing the prevalence and associated factors of overweight/obesity among women of reproductive age in low and middle-income countries with high maternal mortality.

## Materials and methods

### Study design, setting, and period

The Demographic and Health Surveys (DHS) are comprehensive cross-sectional surveys that gather data to provide population and health indicators at both the national and sub-national levels. For this study, we specifically focused on low and middle-income countries having high maternal mortality and available DHS data, which are Benin, Burundi, Chad, Democratic Republic of the Congo (DRC), Cote d'Ivoire, Cameroon, Gambia, Guinea, Haiti, Kenya, Liberia, Lesotho, Madagascar, Mali, Mauritania, Malawi, Nigeria, Niger, Sierra Leone, Togo, and Zimbabwe during the period from 2012 to 2022.

### Data source and population

We compiled data from Demographic and Health Surveys (DHSs) conducted in twenty-one low and middle-income countries that have high maternal mortality (Table 1). Among these countries, four do not have any available DHS reports, and one does not have publicly accessible DHS data. Additionally, one country does not have data on Body Mass Index (BMI), and another country does not have appropriate BMI data. Conducting a multi-country analysis using DHS survey data is feasible and reliable, as the surveys utilize consistent questionnaires, sampling procedures, data collection methods, and coding [53].

Our study encompassed all women of reproductive age (15–49 years) residing in low and middle-income countries with high maternal mortality who had valid BMI measurements and were alive at the time of the survey. For this analysis, we

**Table 1. Maternal mortality category and year of the survey by the country.**

| Serial Number | Country | Year of DHS Survey | Maternal mortality/100,000 | Category |
|---|---|---|---|---|
| 1. | Benin | 2017−18 | 523 | Very High |
| 2. | Burundi | 2016−17 | 494 | High |
| 3. | Chad | 2014−15 | 1063 | Extremely High |
| 4. | Democratic Republic of the Congo (DRC) | 2013−14 | 547 | Very High |
| 5. | Cote d'Ivoire | 2021 | 480 | High |
| 6. | Cameroon | 2018 | 438 | High |
| 7. | Gambia | 2019−20 | 458 | High |
| 8. | Guinea | 2018 | 553 | Very High |
| 9. | Haiti | 2016−17 | 350 | High |
| 10. | Kenya | 2022 | 530 | Very High |
| 11. | Liberia | 2019−20 | 652 | Very High |
| 12. | Lesotho | 2014 | 566 | Very High |
| 13. | Madagascar | 2021 | 392 | High |
| 14. | Mali | 2018 | 440 | High |
| 15. | Mauritania | 2019−21 | 464 | High |
| 16. | Malawi | 2015−16 | 381 | High |
| 17. | Nigeria | 2018 | 1047 | Extremely high |
| 18. | Niger | 2012 | 441 | High |
| 19. | Siera Leone | 2019 | 443 | High |
| 20. | Togo | 2013−14 | 399 | High |
| 21. | Zimbabwe | 2015 | 357 | High |

*High:300–499; Very High:500–999; Extremely High: > 1000 [54].*

*Data sources* http://www.dhsprogram.com, http://srhr.org/mmr2020, or http://www.who.int/publications/i/item/9789240068759.

utilized the individual record (IR) file, encompassing all women of reproductive age at the time of the survey, excluding those who were pregnant, had given birth in the last five years, or had inappropriate BMI measurements based on DHS guidance. The exclusion of postpartum women is due to ongoing physiological changes, such as adjustments in weight and body composition, fluid retention, and hormonal fluctuations that can affect body weight and make BMI an unreliable measure during this period. Besides, another reason for excluding postpartum women is that residual pregnancy-related changes and central fat accumulation may lead to misleading BMI results. We recalculated the sampling weights specifically for the sub-sample of women with valid BMI data, excluding those categorized as underweight. The original DHS weights were designed to apply to the full survey population, but because not all women had BMI data, and our focus was on women with normal, overweight, or obese BMI levels, recalculating the weights ensured that our sub-sample remained representative of the population of interest. Consequently, the total weighted sample size from the combined data was 64,076 (Fig 1).

## Variables and measurements

**Outcome variable.** BMI was the primary variable of interest and determined by dividing weight in kilograms by the square of height in meters [55]. The DHS findings uncovered that measurements were conducted on women of reproductive age (15−49 years old) in all selected households. Weight measurements were taken using portable SECA mother-infant scales with a digital display, which were developed and produced with close oversight from UNICEF. Height measurements were obtained using the Shorr measuring board. We categorized and standardized the BMI (dependent variable) into normal and overweight/obesity as per the WHO classification [56] and the guide to DHS statistics, DHS-7 [55]. Accordingly, a normal BMI is within the range of 18.5 to 24.9 kg/m$^2$, while a BMI ≥ 25 kg/m$^2$ indicates overweight, and those with a BMI ≥ 30 kg/m$^2$ were classified as obese. For analytical purposes, women of reproductive age who were overweight or obese were categorized as "yes" and those with a normal BMI were categorized as "no". We excluded underweight (mildly, moderately, and severely thin) women from both the numerator and denominator, as our focus was specifically on exploring the prevalence and associated factors of overweight/obesity as outlined in the study's topic. The decision to exclude underweight women from this analysis is also supported by previously published articles [41,57].

**Independent variables.** In our analysis, the independent variables consist of individual-level factors such as the age, educational status, occupation, and marital status of women, as well as the number of household members, wealth index, use of contraceptive methods, smoking habits, sex of the household head, number of living children, and access to media sources like television, newspapers/magazines, radio, and the internet.

Additionally, community-level factors, including place of residence, distance to health facilities, community poverty level, community women's education, and community media usage, were also considered. Of the community-level factors, residence (rural, urban) and distance to health facilities (big problem, not a big problem) were directly accessed from the DHS dataset. However, the aggregated community-level independent variables, community-level poverty, community-level media exposure, and community-level women's education were constructed by aggregating individual-level characteristics at the cluster level (i.e., community level). For each community-level variable, we calculated the proportion of women within each cluster who exhibited the characteristic of interest. Specifically, community-level poverty was determined by the proportion of women categorized in the lowest wealth quintiles (poorest and poorer), while community-level media exposure was based on the proportion of women with access to at least one form of media (television, radio, newspapers, or internet). Additionally, community-level women's education was calculated as the proportion of women who had completed at least primary education. These variables were then categorized as "high" or "low" based on the distribution of the computed proportion values for each community. The distribution of these variables was assessed using histograms, and since the aggregate variables were not normally distributed, the median value was used as the cut-off point for categorization [58,59]

 

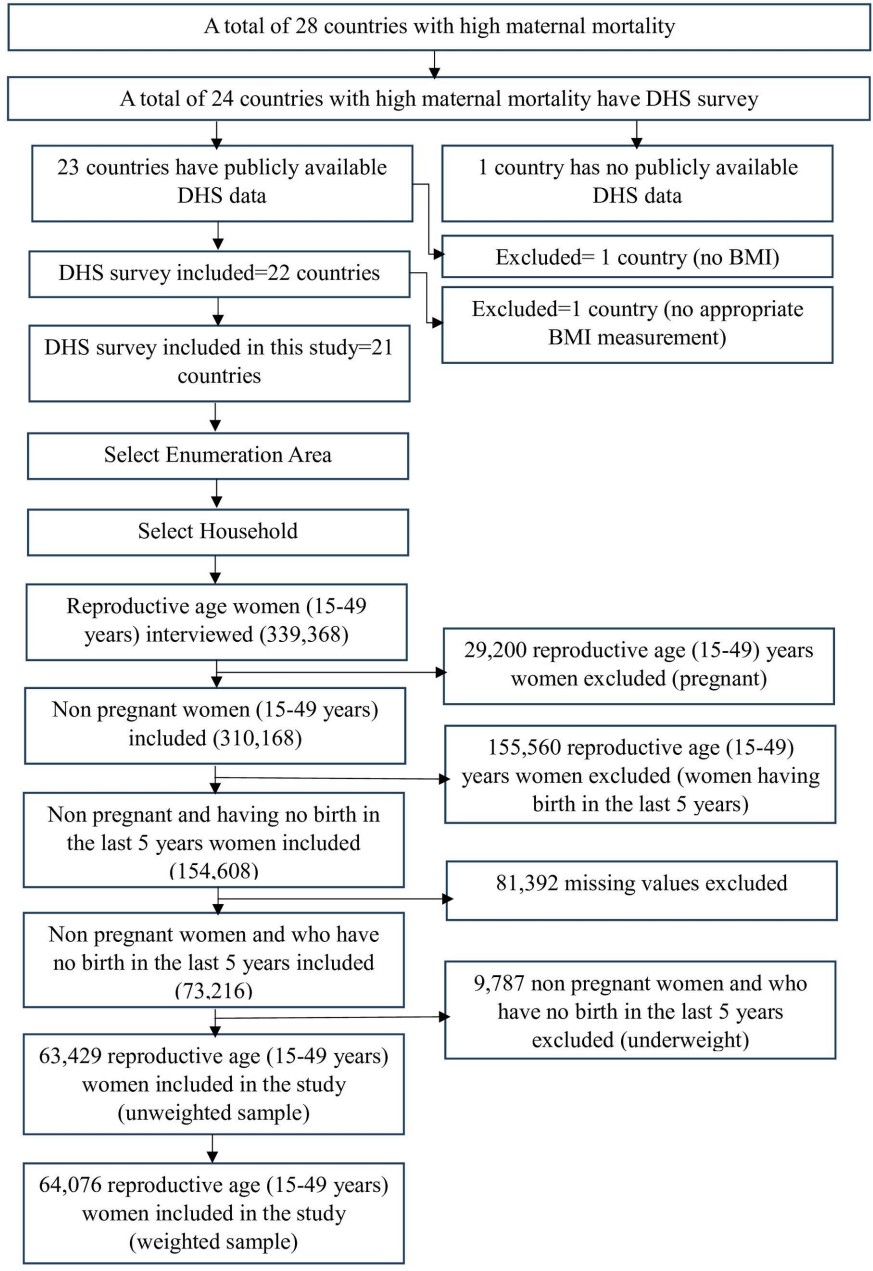

**Fig 1. Flow chart showing the sampling procedure of the DHS data of women of reproductive age in countries with high maternal mortality.**

**Community-level women's education** refers to the percentage of women of reproductive age in the community who have attained primary education or higher. This measure is categorized as low if less than 50% of women have primary education or higher, and high if more than 50% of women have attained this level of education [58–61].

**Community poverty:** Communities were categorized as having high or low economic status based on the proportion of women in the poorest wealth quintiles. To elaborate more, it was categorized as high community economic status if the proportion was less than 50%, whereas it is considered as low community economic status if the proportion was higher than 50% based median value of the aggregated poor wealth status [58–61].

**Community-level media usage** refers to the proportion of women of reproductive age in the community who watched television, listened to the radio, read newspapers/magazines, and used the internet. It was considered low if less than 50% of respondents had media exposure, and high if 50% or more had media exposure [58–61].

**The maternal mortality ratio (MMR)** is categorized as high if it falls within the range of 300–499, very high if it is between 500–999, and extremely high if it equals or exceeds 1000 maternal deaths per 100,000 live births [54]. The focus on countries with high, very high, or extremely high maternal mortality rates is due to the urgent need to address the causes of morbidity and mortality among women in these regions.

### Ethical approval and consent to participate

We have obtained permission from the DHS program to access the data online at http://www.dhsprogram.com and to download and use the data freely. The original collection of DHS data adhered to international and national ethical guidelines, with ethical clearance approved by the ICF Macro Institutional Review Board, the Centers for Disease Control and Prevention (CDC), and the Institutional Review Board (IRB) in each country, per the United States Department of Health and Human Services requirements for human subject protection. Written consent was obtained from all respondents and/or their legal guardians for minors (aged below 16) to participate, in line with Helsinki declarations. Importantly, the procedures for DHS public-use datasets ensure that respondents, households, or sample communities cannot be identified in any way, as the data files do not contain personal identifiers of individuals or household addresses. Furthermore, this study is not an experimental study. For more details on the ethical standards and data usage by the DHS, please refer to: http://goo.gl/ny8T6X.

### Statistical analyses

Data extraction, coding, and analysis were performed using STATA software version 14.2. As the outcome variable was not equally distributed across each cluster, we weighted the sample to ensure its representativeness and alignment with the actual population of each country. Given the hierarchical nature of the DHS, multilevel analysis was conducted to identify significantly associated factors. To assess the variability across the 21 countries, we carried out the calculation of the Intraclass Correlation Coefficient (ICC) and Proportional Change in Variance (PCV). The Likelihood Ratio test (LR) and Median Odds Ratio (MOR) were computed to assess the variation between clusters. Variables with a p-value ≤0.25 in the bi-variable analysis were included in the multivariable analysis, and four models were constructed. These models included the null model (with no predictors), model I (adjusted for individual-level variables only), model II (adjusted for community-level variables only), and model III (adjusted for both individual and community-level variables simultaneously). The nested models were compared using deviance (−2 log-likelihood). Adjusted Odds Ratios (AOR) with 95% confidence intervals and $p < 0.05$ were used to identify significant predictors. The variance inflation factor (VIF) test was conducted to check for multicollinearity and found no issues, as all variables had VIF < 5, with model III VIF at 1.74.

## Results

### Socio-demographic, economic, and health service-related characteristics of respondents

The study included a total of 64,076 women of reproductive age, with 47.09% falling into the 15–24 age group. In this study, the highest number (12.53%) of the study participants were from Kenya, whereas the lowest number (2.22%) of the participants were from Liberia. Approximately half of the participants (50.29%) had attended a secondary education or higher, and about 30.47% were categorized as the richest. Besides, 63.12% of the household heads were female, and 53.04% of the respondents lived in rural areas. Furthermore, 76.11% of the respondents had access to an improved source of drinking water (Tables 2 and 3).

**Table 2. Country-specific data of respondents in low and middle-income countries with high maternal mortality.**

| Variables | Un-weighted frequency (%) | Weighted frequency (%) | Dependent variable: Over-weight/Obesity (weighted) | |
|---|---|---|---|---|
| | | | Yes, n (%) | No, n (%) |
| **Country** | | | | |
| Benin | 2,586 (4.10) | 2,584 (4.03) | 818 (31.7) | 1,766 (68.3) |
| Burundi | 3,117 (4.91) | 3,008 (4.70) | 311 (10.3) | 2,697 (89.7) |
| Democratic Republic of the Congo (DRC) | 2,751 (4.34) | 2,906 (4.53) | 575 (19.8) | 2,331 (80.2) |
| Cote d'Ivoire | 2,927 (4.61) | 3,027 (4.72) | 1,127 (37.2) | 1,900 (62.8) |
| Cameroon | 3,048 (4.81) | 3,000 (4.68) | 1,151 (38.4) | 1,849 (61.6) |
| Gambia | 2,361 (3.72) | 2,540 (4.00) | 1,013 (39.9) | 1,527 (60.1) |
| Guinea | 2,044 (3.22) | 2,053 (3.20) | 621 (30.2) | 1,433 (69.8) |
| Haiti | 5,213 (8.21) | 5,306 (8.28) | 1,817 (34.2) | 3,489 (65.8) |
| Kenya | 7,599 (11.98) | 8,025 (12.53) | 3,311 (41.3) | 4,714 (58.7) |
| Liberia | 1,495 (2.36) | 1,425 (2.22) | 571 (40.1) | 854 (59.9) |
| Lesotho | 1,798 (2.83) | 1,784 (2.78) | 812 (45.5) | 972 (54.5) |
| Madagascar | 3,528 (5.56) | 3,570 (5.57) | 649 (18.2) | 2,921 (81.8) |
| Mali | 1,542 (2.43) | 1,451 (2.26) | 479 (33.0) | 972 (67.0) |
| Mauritania | 3,180 (5.01) | 3,181 (4.96) | 1,691 (53.2) | 1,490 (46.8) |
| Malawi | 2,981 (4.70) | 2,916 (4.55) | 712 (24.4) | 2,204 (75.6) |
| Nigeria | 3,879 (6.12) | 3,783 (5.90) | 1,156 (30.6) | 2,627 (69.4) |
| Niger | 1,131 (1.78) | 1,023 (1.60) | 220 (21.5) | 803 (78.5) |
| Siera Leone | 3,285 (5.18) | 3,373 (5.26) | 1,046 (31.0) | 2,327 (69.0) |
| Chad | 2,837 (4.47) | 2,981 (4.65) | 495 (16.6) | 2,486 (83.4) |
| Togo | 1,941 (3.06) | 2,038 (3.18) | 716 (35.2) | 1,322 (64.8) |
| Zimbabwe | 4,186 (6.60) | 4,100 (6.40) | 1,516 (37.0) | 2,584 (63.0) |

### Prevalence of overweight/obesity among women of reproductive age in low and middle-income countries with high maternal mortality

The pooled prevalence of overweight/obesity among women of reproductive age in low and middle-income countries with high maternal mortality was 32% (95% CI: 27–37), with the highest in Mauritania (53%) and the lowest in Burundi (10%) (Fig 2).

### Multilevel logistic regression analysis of overweight/obesity among women of reproductive age in low and middle-income countries with high maternal mortality

The random effects analysis revealed that the Inter Cluster Correlation Coefficient (ICC) of the null model was 0.19, indicating that nineteen percent of the total variability in the prevalence of overweight/obesity was due to between-cluster variability, while eighty-one percent was due to individual differences. The value of this ICC indicated moderate to strong clustering, suggesting that multilevel or clustered analysis is appropriate. The MOR in the null model was 2.30, suggesting that a woman of reproductive age from a cluster with high overweight/obesity prevalence has a 2.30 times higher chance of being affected than a woman from a cluster with a lower prevalence. The best-fitted model was Model III, which had the highest log likelihood (−27625) and the lowest deviance (55250) among all fitted models. Model III had a Proportional Change in Variance (PCV) of 29%, indicating that 29% of the total variability in overweight/obesity prevalence was explained by the full model (Table 4).

**Table 3. Socio-demographic and health service-related characteristics of respondents in low and middle-income countries with high maternal mortality.**

| Variables | Un-weighted frequency (%) | Weighted frequency (%) | Dependent variable: Overweight/Obesity (weighted) | | | | P value |
|---|---|---|---|---|---|---|---|
| | | | Yes | | No | | |
| | | | n (%) | 95% CI | n (%) | 95% CI | |
| **Age of women** | | | | | | | |
| 15–24 | 29,884 (47.11) | 30,174 (47.09) | 5,045 (16.7) | (16.3,17.1) | 25,129 (83.3) | (82.9,83.7) | <0.001 |
| 25–34 | 10,057 (15.86) | 10,359 (16.17) | 4,472 (43.2) | (42.2,44.1) | 5,887 (56.8) | (55.9,57.8) | |
| 35–49 | 23,488 (37.03) | 23,542 (36.74) | 11,289 (48.0) | (47.3,48.6) | 12,253 (52.0) | (51.4,52.7) | |
| **Women's educational status** | | | | | | | |
| No education | 15,492 (24.42) | 14,766 (23.04) | 4,515 (30.6) | (29.8,31.3) | 10,250 (69.4) | (68.7,70.2) | <0.001 |
| Primary education | 17,064(26.90) | 17,085 (26.66) | 5,548 (32.5) | (31.8,33.2) | 11,537 (67.5) | (66.8,68.2) | |
| Secondary and above | 30,873 (48.67) | 32,225 (50.29) | 10,744 (33.3) | (32.8,33.9) | 21,481 (66.7) | (66.1,67.2) | |
| **Women's occupation** | | | | | | | |
| Not working | 24,520 (40.36) | 24,509 (39.81) | 6,171 (25.2) | (24.6,25.7) | 18,338 (74.8) | (74.3,75.4) | <0.001 |
| Working | 36,235 (59.64) | 37,058 (60.19) | 11,020 (37.8) | (37.3,38.3) | 23,038 (62.2) | (61.7,62.7) | |
| **Women's marital status** | | | | | | | |
| Single | 31,000 (48.87) | 31,481 (49.13) | 6,174 (19.6) | (19.2,20.1) | 25,307 (80.4) | (79.9,80.8) | <0.001 |
| Married | 25,492 (40.19) | 25,549 (39.87) | 11,441 (44.8) | (44.2,45.4) | 14,108 (55.2) | (54.6,55.8) | |
| Widowed | 2,598 (4.10) | 2,579 (4.02) | 1,064 (41.2) | (39.4,43.2) | 1,515 (58.8) | (56.8,60.6) | |
| Divorced | 4,339 (6.84) | 4,467 (6.97) | 2,129 (47.6) | (46.2,49.1) | 2,339 (52.4) | (50.9,53.8) | |
| **Number of household members** | | | | | | | |
| 1-4 | 20,732 (32.68) | 21,214 (33.11) | 7,886 (37.2) | (36.5,37.8) | 13,328 (62.8) | (62.2,63.5) | <0.001 |
| 5-10 | 35,455 (55.90) | 35,640 (55.62) | 10,799 (30.3) | (29.8,30.8) | 24,841 (69.7) | (69.2,70.2) | |
| >10 | 7,242 (11.42) | 7,221 (11.27) | 2,122 (29.4) | (28.3,30.4) | 5,099 (70.6) | (69.6,71.7) | |
| **Wealth index** | | | | | | | |
| Poorest | 8,757 (13.81) | 7,785 (12.15) | 1,322 (17.0) | (16.2,17.8) | 6,464 (83.0) | (82.2,83.8) | <0.001 |
| Poorer | 9,976 (15.73) | 9,756 (15.23) | 2,143 (22.0) | (21.2,22.8) | 7,613 (78.0) | (77.2,78.8) | |
| Middle | 12,575 (19.83) | 12,108 (18.90) | 3,465 (28.6) | (27.8,29.4) | 8,643 (71.4) | (70.6,72.2) | |
| Richer | 14,567 (22.97) | 14,906 (23.26) | 5,336 (35.8) | (35.0,36.6) | 9,570 (64.2) | (63.4,65.0) | |
| Richest | 17,554 (27.68) | 19,521 (30.47) | 8,541 (43.8) | (43.1,44.4) | 10,980 (56.2) | (55.6,56.9) | |
| **Current contraceptive method** | | | | | | | |
| No method | 50,638 (79.83) | 50,821 (79.31) | 15,039 (29.6) | (29.2,30.0) | 35,782 (70.4) | (70.0,70.8) | <0.001 |
| Modern method | 11,360 (17.91) | 11,678 (18.22) | 5,100 (43.7) | (42.8,44.6) | 6,578 (56.3) | (55.4,57.2) | |
| Traditional method | 1,431 (22.56) | 1,577 (2.46) | 668 (42.4) | (40.0,44.6) | 908 (57.6) | (55.2,60.0) | |
| **Smoking** | | | | | | | |
| No | 60,043 (99.1) | 60,512 (99.10) | 20,073 (33.2) | (32.8,33.5) | 40,439 (66.8) | (66.5,67.2) | <0.01 |
| Yes | 549 (0.90) | 584 (0.9) | 240 (41.0) | (37.1,45.0) | 344 (59.0) | (55.0,62.9) | |
| **Sex of household head** | | | | | | | |
| Male | 40,184 (63.35) | 40,448 (63.12) | 12,575 (31.1) | (30.6,31.5) | 27,873 (68.9) | (68.5,69.4) | <0.001 |
| Female | 23,245 (36.65) | 23,628 (36.88) | 8,232 (34.8) | (34.2,35.5) | 15,396 (65.2) | (64.5,65.8) | |
| **Number of living children** | | | | | | | |
| No child | 34,455 (54.32) | 34,906 (54.48) | 7,115 (20.4) | (20.0,20.8) | 27,791 (79.6) | (79.2,80.0) | <0.001 |
| One child | 4,913 (7.75) | 5,163 (8.10) | 2,415 (46.8) | (45.4,48.1) | 2,748 (53.2) | (51.9,54.6) | |
| 2-3 children | 10,502 (16.56) | 10,792 (16.84) | 5,592 (51.8) | (50.9,52.8) | 5,200 (48.2) | (47.1,49.1) | |
| ≥ 4 children | 13,559 (21.38) | 13,215 (20.62) | 5,685 (43.0) | (42.2,43.9) | 7,530 (57.0) | (56.1,57.8) | |

*(Continued)*

**Table 3.** (Continued)

| Variables | Un-weighted frequency (%) | Weighted frequency (%) | Dependent variable: Overweight/Obesity (weighted) | | | | |
|---|---|---|---|---|---|---|---|
| | | | Yes | | No | | P value |
| | | | n (%) | 95% CI | n (%) | 95% CI | |
| **Frequency of watching television** | | | | | | | |
| Not at all | 31,491 (49.68) | 30,584 (47.77) | 7,236 (23.7) | (23.2,24.1) | 23,348 (76.3) | (75.9,76.8) | <0.001 |
| Less than once a week | 10,254 (16.18) | 10,236 (15.99) | 3,600 (35.2) | (34.2,36.1) | 6,636 (64.8) | (63.9,65.8) | |
| At least once a week | 21,641 (34.14) | 23,207 (36.25) | 9,964 (42.9) | (42.3,43.6) | 13,244 (57.1) | (56.4,57.7) | |
| **Frequency of reading a newspaper or a magazine** | | | | | | | |
| Not at all | 49,409 (77.93) | 49,253 (76.91) | 15,261 (31.0) | (30.6,31.4) | 33,993 (69.0) | (68.6,69.4) | <0.001 |
| Less than once a week | 8,305 (13.10) | 8,662 (13.53) | 3,191 (36.8) | (35.8,37.9) | 5,471 (63.2) | (62.1,64.2) | |
| At least once a week | 5,690 (8.97) | 6,123 (9.56) | 2,347 (38.3) | (37.1,39.6) | 3,776 (61.7) | (60.4,62.9) | |
| **Frequency of listening to the radio** | | | | | | | |
| Not at all | 25,681 (40.49) | 24,937 (38.92) | 6,697 (26.9) | (26.3,27.4) | 18,240 (73.1) | (72.6,73.7) | <0.001 |
| Less than once a week | 13,880 (21.88) | 14,220 (22.19) | 4,796 (33.7) | (33.0,34.5) | 9,425 (66.3) | (65.5,67.0) | |
| At least once a week | 23,868 (37.63) | 24,918 (38.89) | 9,314 (37.4) | (36.8,38.0) | 15,604 (62.6) | (62.0,63.2) | |
| **Frequency of using the internet** | | | | | | | |
| Not at all | 39,557 (74.68) | 38,666 (72.48) | 11,261 (29.1) | (28.7,29.6) | 27,405 (70.9) | (70.4,71.3) | <0.001 |
| Less than once a week | 1,496 (2.82) | 1,570 (2.94) | 616 (39.2) | (36.8,41.6) | 955 (60.8) | (58.4,63.2) | |
| At least once a week | 11,918 (22.50) | 13,107 (24.57) | 6,112 (46.6) | (45.8,47.5) | 6,995 (53.4) | (52.5,54.2) | |
| **Source of drinking water** | | | | | | | |
| Unimproved | 15,970 (25.18) | 15,304 (23.88) | 3,538 (23.1) | (22.5,23.8) | 11,766 (76.9) | (76.2,77.5) | <0.001 |
| Improved | 47,456 (74.82) | 48,769 (76.11) | 17,267 (35.4) | (35.0,35.8) | 31,502 (64.6) | (64.2,65.0) | |
| **Residence** | | | | | | | |
| Urban | 28,811 (45.42) | 30,088 (46.96) | 12,432 (41.3) | (40.8,41.9) | 17,656 (58.7) | (58.1,59.2) | <0.001 |
| Rural | 34,618 (54.58) | 33,988 (53.04) | 8,375 (24.6) | (24.2,25.1) | 25,613 (75.4) | (74.9,75.8) | |
| **Distance to health facility** | | | | | | | |
| Big problem | 19,736 (32.57) | 19,224 (31.47) | 5,344 (27.8) | (27.2,28.4) | 13,880 (72.2) | (71.6,72.8) | <0.001 |
| Not a big problem | 40,854 (67.43) | 41,869 (68.53) | 14,968 (35.8) | (35.3,36.2) | 26,901 (64.2) | (63.8,64.7) | |
| **Community-level women education** | | | | | | | |
| Low | 32,068 (50.56) | 31,242 (48.76) | 8,861 (28.4) | (27.9,27.9) | 22,381 (71.6) | (71.1,72.1) | <0.001 |
| High | 31,361 (49.44) | 32,833 (51.24) | 11,946 (36.4) | (35.9,36.9) | 20,887 (63.6) | (63.1,64.1) | |
| **Community poverty** | | | | | | | |
| High community economic status | 31,633 (49.87) | 33,598 (52.43) | 13,613 (40.5) | (40.0,41.0) | 19,985 (59.5) | (59.0,60.0) | <0.001 |
| Low community economic status | 31796 (50.13) | 30478 (47.56) | 7,193 (23.6) | (23.1,24.1) | 23,284 (76.4) | (75.9,76.9) | |
| **Community-level media usage** | | | | | | | |
| Low | 26,760 (50.52) | 25,531 (47.86) | 6,757 (26.5) | (25.9,27.0) | 18,774 (73.5) | (73.0,74.1) | <0.001 |
| High | 26,211 (49.48) | 27,812 (52.14) | 11,231 (40.4) | (39.8,41.0) | 16,581 (59.6) | (59.0,60.2) | |

In a study aiming to identify the factors contributing to overweight/obesity among women of reproductive age in low and middle-income countries with high maternal mortality, a multivariable analysis was carried out. According to the best-fitted model (Model III), both individual and community-level variables were found to be significant factors in overweight/obesity among women of reproductive age.

The individual-level factors included women's age, women's educational status, women's occupation, women's marital status, as well as households' income level and the sex of the household head. Moreover, the number of children,

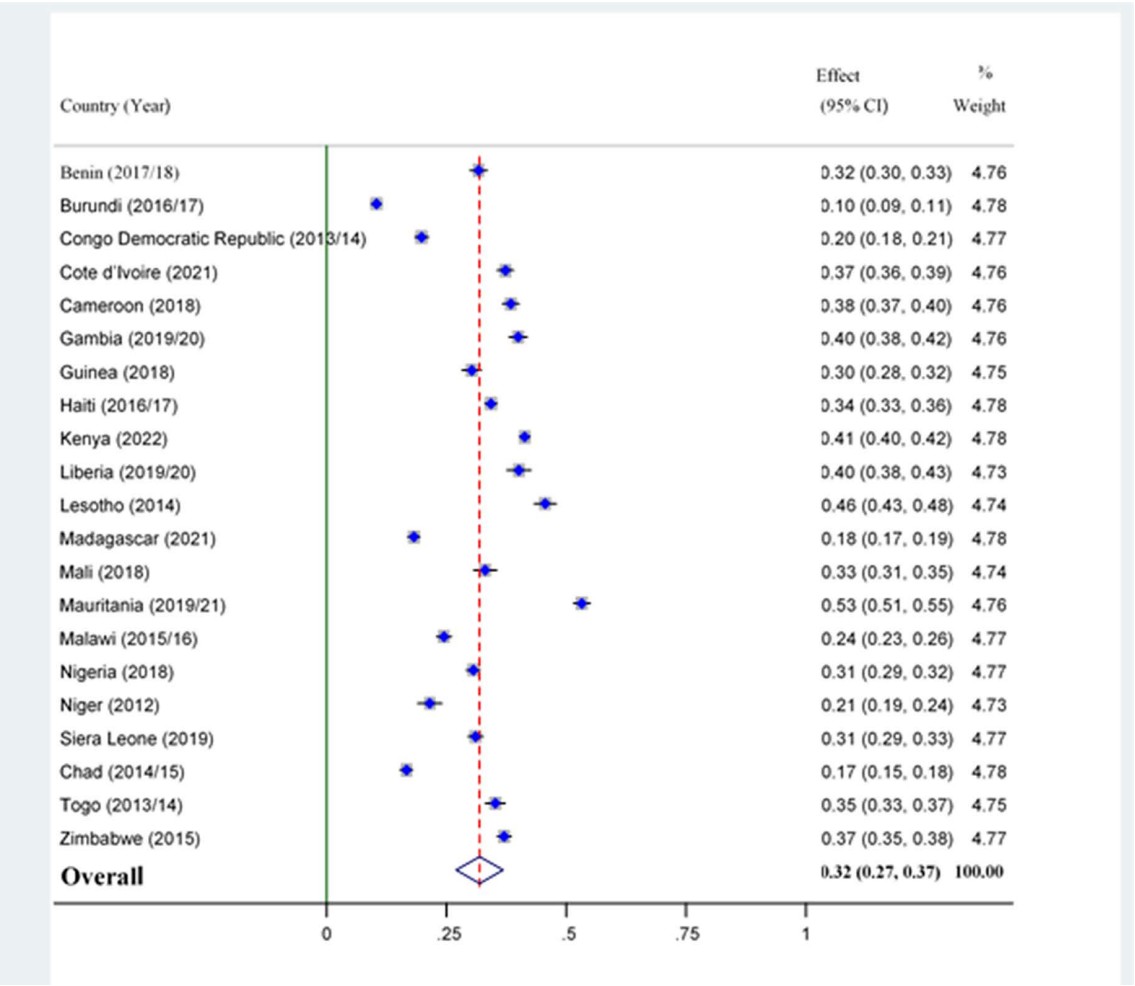

**Fig 2. Prevalence of overweight/obesity of women of reproductive age in countries with high maternal mortality.**

frequency of watching television and internet usage, and sources of drinking water were also identified as individual-level factors.

On the other hand, residence, community poverty, and community-level media usage were found to be significantly associated variables at the community level. These findings are outlined in Table 4.

## Discussion

Overweight/obesity in women of reproductive age (15–49 years old) is a major public health problem in low and middle-income countries. Early detection and prevention of overweight/ obesity can greatly improve the quality of life for women and households. To address this issue, it is crucial to determine the prevalence of overweight/ obesity and identify associated factors. This study assessed the pooled prevalence of overweight/obesity and its associated factors among women of reproductive age (15–49 years old) in low and middle-income countries with high maternal mortality using recent DHS surveys. Our research revealed that 32% of women of reproductive age are overweight/obese, indicating that more than three out of ten women fall into this category. The study found that Mauritania has the highest

**Table 4. Multivariable multilevel logistic regression analysis results of both individual-level and community-level factors associated with overweight/obesity in low and middle-income countries with high maternal mortality.**

| Variables | Null model | Model I AOR [95% CI] | P-value (Model I) | Model II AOR [95% CI] | P-value (Model II) | Model III AOR [95% CI] | P-value (Model III) |
|---|---|---|---|---|---|---|---|
| **Age of women** | | | | | | | |
| 15-24 | | 1.00 | | | | 1.00 | |
| 25-34 | | 2.44 (2.25 - 2.65) | 0.000 | | | 2.42 (2.23 - 2.63) | 0.000 |
| 35-49 | | 3.49 (3.17 - 3.84) | 0.000 | | | 3.47 (3.15 - 3.83) | 0.000 |
| **Women's educational status** | | | | | | | |
| No education | | 1.00 | | | | 1.00 | |
| Primary education | | 1.10 (1.03 - 1.18) | 0.004 | | | 1.10 (1.02 - 1.18) | 0.012 |
| Secondary and above | | 1.10 (1.02 - 1.19) | 0.011 | | | 1.10 (1.01 - 1.18) | 0.035 |
| **Women's occupation** | | | | | | | |
| Not working | | 1.07 (1.02 - 1.13) | 0.012 | | | 1.06 (1.01–1.12) | 0.026 |
| Working | | 1.00 | | | | 1.00 | |
| **Women's marital status** | | | | | | | |
| Single | | 1.00 | | | | 1.00 | |
| Married | | 1.96 (1.81 - 2.13) | 0.000 | | | 1.98 (1.82 - 2.15) | 0.000 |
| Widowed | | 1.39 (1.22 - 1.59) | 0.000 | | | 1.39 (1.22 - 1.59) | 0.000 |
| Divorced | | 1.71 (1.54 −1.90) | 0.000 | | | 1.70 (1.53 −1.88) | 0.000 |
| **Number of household members** | | | | | | | |
| 1–4 | | 1.00 | | | | 1.00 | |
| 5–10 | | 1.00 (0.95 - 1.05) | 0.896 | | | 1.00 (0.95 - 1.05) | 0.977 |
| >10 | | 1.03 (0.94 - 1.12) | 0.566 | | | 1.03 (0.94 - 1.12) | 0.562 |
| **Wealth index** | | | | | | | |
| Poorest | | 1.00 | | | | 1.00 | |
| Poorer | | 1.33 (1.21 - 1.47) | 0.000 | | | 1.30 (1.18 - 1.44) | 0.000 |
| Middle | | 1.93 (1.75 - 2.13) | 0.000 | | | 1.75 (1.59 - 1.94) | 0.000 |
| Richer | | 2.44 (2.21 −2.70) | 0.000 | | | 1.98 (1.77 - 2.21) | 0.000 |
| Richest | | 3.16 (2.84 - 3.52) | 0.000 | | | 2.38 (2.11 - 2.69) | 0.000 |
| **Current contraceptive method** | | | | | | | |
| No method | | 0.95 (0.89 - 1.00) | 0.069 | | | 0.95 (0.89 −1.00) | 0.061 |
| Modern method | | 1.00 | | | | 1.00 | |
| Traditional method | | 1.13 (0.98 - 1.31) | 0.092 | | | 1.12 (0.97 - 1.30) | 0.118 |
| **Smoking** | | | | | | | |
| No | | 1.00 | | | | 1.00 | |
| Yes | | 0.94 (0.77 - 1.15) | 0.550 | | | 0.93 (0.76 - 1.14) | 0.511 |
| **Sex of household head** | | | | | | | |
| Male | | 1.00 | | | | 1.00 | |
| Female | | 1.18 (1.12 - 1.24) | 0.000 | | | 1.15 (1.09 - 1.22) | 0.000 |
| **Number of living children** | | | | | | | |
| No child | | 1.00 | | | | 1.00 | |
| One child | | 1.28 (1.16 - 1.41) | 0.000 | | | 1.27 (1.15 - 1.40) | 0.000 |
| 2-3 children | | 1.45 (1.31 - 1.59) | 0.000 | | | 1.44 (1.31 - 1.59) | 0.000 |
| ≥ 4 children | | 1.34 (1.20 - 1.49) | 0.000 | | | 1.34 (1.21 - 1.50) | 0.000 |
| **Frequency of watching television** | | | | | | | |
| Not at all | | 1.00 | | | | 1.00 | |
| Less than once a week | | 1.39 (1.30 - 1.50) | 0.000 | | | 1.30 (1.21 - 1.39) | 0.000 |

*(Continued)*

**Table 4.** (Continued)

| Variables | Null model | Model I AOR [95% CI] | P-value (Model I) | Model II AOR [95% CI] | P-value (Model II) | Model III AOR [95% CI] | P-value (Model III) |
|---|---|---|---|---|---|---|---|
| At least once a week | | 1.55 (1.46 - 1.66) | 0.000 | | | 1.41 (1.32 - 1.51) | 0.000 |
| **Frequency of reading a newspaper or a magazine** | | | | | | | |
| Not at all | | 1.00 | | | | 1.00 | |
| Less than once a week | | 1.03 (0.97 - 1.11) | 0.344 | | | 1.04 (0.97 - 1.11) | 0.262 |
| At least once a week | | 1.00 (0.92 - 1.10) | 0.993 | | | 1.00 (0.92 - 1.10) | 0.964 |
| **Frequency of listening to the radio** | | | | | | | |
| Not at all | | 1.00 | | | | 1.00 | |
| Less than once a week | | 0.98 (0.92 - 1.05) | 0.605 | | | 0.97 (0.91 - 1.03) | 0.298 |
| At least once a week | | 0.98 (0.92 - 1.03) | 0.387 | | | 0.97 (0.92 - 1.03) | 0.328 |
| **Frequency of using the internet** | | | | | | | |
| Not at all | | 1.00 | | | | 1.00 | |
| Less than once a week | | 1.50 (1.32 - 1.71) | 0.000 | | | 1.45 (1.28 - 1.65) | 0.000 |
| At least once a week | | 1.74 (1.64 - 1.85) | 0.000 | | | 1.67 (1.57 - 1.78) | 0.000 |
| **Source of drinking water** | | | | | | | |
| Unimproved | | 1.00 | | | | 1.00 | |
| Improved | | 1.12 (1.05 - 1.19) | 0.000 | | | 1.08 (1.01 - 1.15) | 0.020 |
| **Community-level variables** | | | | | | | |
| **Residence** | | | | | | | |
| Urban | | | | 1.48 (1.38 - 1.58) | 0.000 | 1.38 (1.28 - 1.49) | 0.000 |
| Rural | | | | 1.00 | | 1.00 | |
| **Distance to health facility** | | | | | | | |
| Big problem | | | | 1.00 | | 1.00 | |
| Not a big problem | | | | 1.12 (1.07 - 1.18) | 0.000 | 1.04 (0.99 - 1.10) | 0.107 |
| **Community-level women's education** | | | | | | | |
| Low | | | | 1.00 | | 1.00 | |
| High | | | | 1.09 (1.03 - 1.45) | 0.003 | 1.02 (0.95 - 1.09) | 0.598 |
| **Community poverty** | | | | | | | |
| High community economic status | | | | 1.52 (1.42 - 1.63) | 0.000 | 1.10 (1.01 - 1.20) | 0.032 |
| Low community economic status | | | | 1.00 | | 1.00 | |
| **Community-level media usage** | | | | | | | |
| Low | | | | 1.00 | | 1.00 | |
| High | | | | 1.46 (1.38 - 1.55) | 0.000 | 1.20 (1.12 - 1.29) | 0.000 |
| **Random effect** | | | | | | | |
| Variance | 0.77 | 0.56 | | 0.48 | | 0.55 | |
| ICC | 0.19 | 0.15 | | 0.13 | | 0.14 | |
| MOR | 2.30 | 2.04 | | 1.93 | | 2.02 | |
| PCV | Reff | 27.30 | | 38.00 | | 29.00 | |
| **Model comparison** | | | | | | | |
| Log likelihood ratio | −38895 | −27706 | | −32282 | | −27625 | |
| Deviance | 77790 | 55412 | | 64564 | | 55250 | |
| Mean VIF | | 1.64 | | 1.40 | | 1.74 | |

The association is statistically significant at P-value < 0.05.

*ICC = Inter-cluster correlation coefficient, MOR = Median odds ratio, PCV = proportional change in variance. AOR = adjusted odds ratio; CI = confidence interval, VIF = variance inflation factor.*

prevalence of overweight/obesity at 53%, while Burundi has the lowest at 10%. These disparities among countries might be attributed to variations in socio-demographic and economic factors, lifestyle differences such as food preferences, cultural beliefs regarding dietary habits, and disparities in government and policymakers' efforts to address overweight/obesity [20,62,63].

This study found that both individual-level and community-level factors were significantly associated with overweight/obesity in women of reproductive age. These factors include older age, higher education, better economic status, having more living children, frequent television watching, frequent internet usage, not being part of the workforce, marital status, female household head, access to improved drinking water, urban residence, higher community economic status, and higher community-level media usage. These factors were associated with a higher prevalence of overweight/obesity.

As observed in previous research conducted in various settings [14,19–21,27,41,43], our study also found that women aged 25–34 years and 35–49 years were more likely to experience weight gain. This positive association might be attributed to the tendency of BMI to increase with age, even in premenopausal women. This trend is likely driven by age-related physiological changes, including hormonal shifts, increased body fat, and alterations in body composition [41]. Additionally, reduced participation in physical activities due to age-related physiological changes among women may contribute to the increased weight gain [64–66]. Moreover, this might be related to the natural decline in basal metabolic rate with age. As individuals grow older, their metabolism slows down, resulting in reduced energy expenditure at rest. If caloric intake remains unchanged, this can contribute to gradual weight gain over time. Furthermore, the shifts in behavior, lifestyle, and even emotional health might be related to weight gain in older age. This finding, which is supported by the strong odds ratio, indicates that older women are highly vulnerable to weight gain, highlighting the importance of closely monitoring and supporting this population group to prevent obesity.

This study has established a clear association between women's education and overweight/obesity in low and middle-income countries with high maternal mortality. It was found that women of reproductive age with primary, secondary, and higher education were more likely to be overweight/obese compared to those with no education. This result is consistent with previous studies conducted in other locations [7,9,18,41]. The reason behind this trend could be that educated women tend to have higher incomes, leading to changes in lifestyle such as dietary habits and consumption of calorie-dense foods and high-fat diets [4,41].

However, it is important to note that the findings of educational status in our study differ from some other previous studies [67,68]. This variation could be attributed to differences in research methodologies and the datasets used. Our study utilized non-pregnant women's data from the Demographic and Health Surveys (DHS) of 21 countries, with nutritional status measured by BMI. In contrast, the aforementioned studies used BMI data from pregnant women, which might not be the recommended method for assessing nutritional status. In clinical practice, the recommended assessment modality for pregnant women's nutritional status is Mid-Upper Arm Circumference (MUAC), not BMI. Additionally, our study excluded pregnant women in line with the DHS-7 guidelines to ensure reliable estimates. Another potential reason for the disparity in findings could be the differences in the populations studied. Our research focused on developing countries, while the other studies were conducted in more developed nations. It is plausible that women in developed countries might leverage their advanced education to manage weight gain better.

Not working women of reproductive age have been found to have higher odds of overweight/obesity compared to their working counterparts. This aligns with findings from studies conducted in Ethiopia and Bangladesh [15,43]. This could be attributed to lower levels of physical activity related to occupation among non-working women, leading to weight gain [69,70]. Besides, the association between not working women and higher frequencies of television watching or internet use may also contribute to this trend. Previous studies have indicated a strong link between television watching and a sedentary lifestyle, ultimately resulting in increased weight gain [20,71]. Consequently, not working women of reproductive age might have less exposure to physical activity due to the absence of work, potentially increasing their likelihood of gaining weight.

In low and middle-income countries with high maternal mortality, this research indicates a significant correlation between individual-level factors such as frequent television watching and internet usage, as well as community-level media exposure, and the prevalence of overweight/obesity among women of reproductive age. The findings align with similar studies conducted in Mali [20] and Timor-Leste [17]. It is suggested that high levels of television viewing, internet usage, and community media exposure may contribute to abnormal metabolism associated with a sedentary lifestyle [71], potentially leading to increased weight gain [72]. Additionally, the frequent use of media could potentially replace the time that would otherwise be spent on physical activity, resulting in lower energy expenditure and subsequent weight gain [73]. Moreover, internet usage might be linked with higher socio-economic grouping, which in turn leads to higher fat and energy-dense foods that could finally result in weight gain among internet-utilizing women [74]. Moreover, internet usage might serve as a proxy for greater wealth and lifestyle behaviors, which in turn could have a direct effect on weight gain. The odds ratio in this study demonstrated a strong association between media usage and weight gain; therefore, it is crucial to promote digital wellness strategies that encourage regular breaks, physical activity, and healthy eating practices.

The study findings reveal a direct association between economic status and overweight/obesity. In developing countries, higher household income is associated with an increased likelihood of weight gain, as evidenced by the rising and strong odds ratios. Additionally, higher community economic status is significantly associated with higher odds of weight gain in women of reproductive age. These findings are consistent with several previous studies [8,14,15,18–22,40,41,43–45]. One possible explanation for this is that in developing countries, individuals with higher wealth are more inclined to consume energy-dense foods and lead sedentary lifestyles [40,74]. Moreover, women in higher wealth quantiles might have a higher caloric intake compared to those in lower quantiles, which could be a primary factor contributing to weight gain. The finding of our study implies that women of reproductive age from the wealthiest segments in developing countries are disproportionately affected by weight gain, emphasizing the need for targeted obesity prevention and management strategies for this group.

Whereas, paradoxically, in developed countries, there was an inverse association between economic status and weight gain, which depicted that lower economic status was related to higher weight gain. The suggested possible justification could be that poor socio-economic groups in developed countries are more likely to be unemployed, less educated, and might have irregular meals [75].

The odds of overweight/obesity among urban women of reproductive age were higher compared with women in rural areas. This finding is supported by previous studies [8,10,16,21,41]. Research has shown that urbanization significantly influences dietary patterns. The presence of fast-food restaurants offering energy-dense foods [74] and supermarkets might contribute to the higher rates of overweight/obesity. Additionally, urbanization tends to decrease physical activity levels as residents often rely on transportation instead of walking. In contrast, rural residents are more exposed to physical activities due to limited transportation options such as taxis [16]. Therefore, for the reasons mentioned above, urban women of reproductive age might be more susceptible to overweight/obesity compared to their rural counterparts. Thus, our study's finding implies that urban residency in developing countries with high maternal mortality is strongly associated with weight gain, underscoring the need to promote physical activity and healthy dietary habits among urban women of reproductive age.

In low and middle-income countries with high maternal mortality, married, widowed, or divorced women were found to have a higher prevalence of overweight/obesity compared to single women. This observation is consistent with previous studies conducted in various settings [8,10,16,19–21]. The possible explanation for this finding is that women who are not single might have a greater likelihood of having had multiple children in previous pregnancies. Consequently, these women might be at a higher risk of weight gain during and after pregnancy due to a more sedentary lifestyle and consumption of energy-dense foods [74,76]. However, one study revealed a controversial finding that divorced or separated women had intentionally greater weight loss in the past years compared to never-married women [77]. Our study also indicated that the odds of overweight/obesity among women of reproductive age who had children were higher than those who had not given birth. This finding aligns with studies conducted in Cambodia [16] and Guinea [45]. The rationale behind this association could be that women with children might have less time for managing their weight due to the demands of childcare

[16]. Furthermore, having more children is associated with multiple pregnancies, which in turn might lead to depression and anxiety. These psychological factors could potentially contribute to abnormal obesity through hypothalamic-pituitary abnormal hyperactivity [78]. The result of our study confirmed that having a higher number of children significantly contributes to obesity and overweight among women of reproductive age, highlighting the need for crucial attention and support for this group.

The study revealed that female heads of households were more likely to be overweight or obese compared to other household heads. This could be attributed to the stress, depression, and anxiety experienced by female household heads [75]. These conditions are linked to abnormal hormonal function, which can lead to unhealthy weight gain [27,74]. Additionally, the study found that women of reproductive age with improved sources of drinking water had higher odds of being overweight or obese compared to those with unimproved sources. The study highlighted borehole water as a significant component of improved drinking water sources, and this water is known to be salty. Therefore, women with higher salt intake might have a greater likelihood of increased caloric intake, food consumption, and consumption of meat and snacks, which directly contribute to obesity [76]. Another potential explanation could be the presence of bottled water as part of the improved drinking water sources in the study. The study suggested that the chemicals in plastic bottles might contribute to weight gain by altering metabolism and promoting the growth of fat cells. Interestingly, the study confirmed that 94% of women of reproductive age who used bottled water were overweight or obese [77].

Our study had some strengths and limitations. Among the strengths of this study, it was analyzed using pooled data from 21 nationally representative DHS surveys in low and middle-income countries with high maternal mortality. In addition, we weighted our sample size to get reliable estimates and standard errors. As a limitation, the DHS used a cross-sectional survey design, limiting the ability to establish causal relationships between women of reproductive age, overweight/obesity, and the independent variables. The other limitation is the exclusion of underweight women, which may overlook their distinct health challenges. Although BMI is a widely accepted metric for assessing body weight relative to height and is commonly used in the standard DHS statistical guidelines as a key indicator of nutritional status and overall health, it comes with certain limitations and challenges as an indicator of both weight and health. Therefore, future researchers are encouraged to use primary data sources to incorporate more appropriate measurements such as waist circumference, waist-to-hip ratio, and body composition analysis.

## Conclusion and recommendation

The prevalence of overweight/obesity among women of reproductive age (15–49 years old) was found to be relatively high in low and middle-income countries with high maternal mortality. The result indicated that more than three out of ten women in this age group were overweight/obese. Various individual-level and community-level factors were associated with overweight/obesity. Women of reproductive age who were older, more educated, wealthier, had children, watched television, used the internet, had access to improved drinking water sources, and lived in urban areas were more likely to be overweight or obese. Additionally, the odds of being overweight or obese were higher for female heads of households, non-working women, married, widowed, and divorced women. It is recommended to pay special attention to women of reproductive age who have formal education, are not in the workforce, engage in television viewing and internet usage, live in urban areas, are female heads of households, and are among the wealthiest group of women.

## Supporting information

**S1 File. Data used for the analysis of this study.**
(CSV)

## Acknowledgments

The authors would like to thank Measure DHS for their permission to access the dataset.

## Author contributions

**Conceptualization:** Demiss Mulatu Geberu.

**Data curation:** Demiss Mulatu Geberu, Kaleab Mesfin Abera, Yawkal Tsega, Abel Endawkie, Wubshet D. Negash, Amare Mesfin Workie, Lamrot Yohannes, Mihret Getnet, Nigusu Worku, Adina Yeshambel Belay, Lakew Asmare, Hiwot Tadesse Alemu, Misganaw Guadie Tiruneh, Asebe Hagos, Melak Jejaw, Kaleb Assegid Demissie.

**Formal analysis:** Demiss Mulatu Geberu, Kaleab Mesfin Abera, Yawkal Tsega, Abel Endawkie, Wubshet D. Negash, Amare Mesfin Workie, Lamrot Yohannes, Mihret Getnet, Nigusu Worku, Adina Yeshambel Belay, Lakew Asmare, Hiwot Tadesse Alemu, Misganaw Guadie Tiruneh, Asebe Hagos, Melak Jejaw, Kaleb Assegid Demissie.

**Investigation:** Demiss Mulatu Geberu, Kaleab Mesfin Abera, Yawkal Tsega, Abel Endawkie, Wubshet D. Negash, Amare Mesfin Workie, Lamrot Yohannes, Mihret Getnet, Nigusu Worku, Adina Yeshambel Belay, Lakew Asmare, Hiwot Tadesse Alemu, Misganaw Guadie Tiruneh, Asebe Hagos, Melak Jejaw, Kaleb Assegid Demissie.

**Methodology:** Demiss Mulatu Geberu, Kaleab Mesfin Abera, Yawkal Tsega, Abel Endawkie, Wubshet D. Negash, Amare Mesfin Workie, Lamrot Yohannes, Mihret Getnet, Nigusu Worku, Adina Yeshambel Belay, Lakew Asmare, Hiwot Tadesse Alemu, Misganaw Guadie Tiruneh, Asebe Hagos, Melak Jejaw, Kaleb Assegid Demissie.

**Project administration:** Demiss Mulatu Geberu, Kaleab Mesfin Abera, Yawkal Tsega, Abel Endawkie, Wubshet D. Negash, Amare Mesfin Workie, Lamrot Yohannes, Mihret Getnet, Nigusu Worku, Adina Yeshambel Belay, Lakew Asmare, Hiwot Tadesse Alemu, Misganaw Guadie Tiruneh, Asebe Hagos, Melak Jejaw, Kaleb Assegid Demissie.

**Resources:** Demiss Mulatu Geberu, Kaleab Mesfin Abera, Yawkal Tsega, Abel Endawkie, Wubshet D. Negash, Amare Mesfin Workie, Lamrot Yohannes, Mihret Getnet, Nigusu Worku, Adina Yeshambel Belay, Lakew Asmare, Hiwot Tadesse Alemu, Misganaw Guadie Tiruneh, Asebe Hagos, Melak Jejaw, Kaleb Assegid Demissie.

**Software:** Demiss Mulatu Geberu, Kaleab Mesfin Abera, Yawkal Tsega, Abel Endawkie, Wubshet D. Negash, Amare Mesfin Workie, Lamrot Yohannes, Mihret Getnet, Nigusu Worku, Adina Yeshambel Belay, Lakew Asmare, Hiwot Tadesse Alemu, Misganaw Guadie Tiruneh, Asebe Hagos, Melak Jejaw, Kaleb Assegid Demissie.

**Supervision:** Demiss Mulatu Geberu, Kaleab Mesfin Abera, Yawkal Tsega, Abel Endawkie, Wubshet D. Negash, Amare Mesfin Workie, Lamrot Yohannes, Mihret Getnet, Nigusu Worku, Adina Yeshambel Belay, Lakew Asmare, Hiwot Tadesse Alemu, Misganaw Guadie Tiruneh, Asebe Hagos, Melak Jejaw, Kaleb Assegid Demissie.

**Validation:** Demiss Mulatu Geberu, Asebe Hagos.

**Visualization:** Demiss Mulatu Geberu, Kaleab Mesfin Abera, Yawkal Tsega, Abel Endawkie, Wubshet D. Negash, Amare Mesfin Workie, Lamrot Yohannes, Mihret Getnet, Nigusu Worku, Adina Yeshambel Belay, Lakew Asmare, Hiwot Tadesse Alemu, Misganaw Guadie Tiruneh, Asebe Hagos, Melak Jejaw, Kaleb Assegid Demissie.

**Writing – original draft:** Demiss Mulatu Geberu, Kaleab Mesfin Abera, Yawkal Tsega, Abel Endawkie, Wubshet D. Negash, Amare Mesfin Workie, Lamrot Yohannes, Mihret Getnet, Nigusu Worku, Adina Yeshambel Belay, Lakew Asmare, Hiwot Tadesse Alemu, Misganaw Guadie Tiruneh, Asebe Hagos, Melak Jejaw, Kaleb Assegid Demissie.

**Writing – review & editing:** Demiss Mulatu Geberu, Kaleab Mesfin Abera, Yawkal Tsega, Abel Endawkie, Wubshet D. Negash, Amare Mesfin Workie, Lamrot Yohannes, Mihret Getnet, Nigusu Worku, Adina Yeshambel Belay, Lakew Asmare, Hiwot Tadesse Alemu, Misganaw Guadie Tiruneh, Asebe Hagos, Melak Jejaw, Kaleb Assegid Demissie.

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
