## [Decision Letter · Decision Letter 0]

7 Jul 2024

PONE-D-23-42732Pooled prevalence and determinants of overweight/obesity among reproductive age women in countries with high maternal mortality:  A multi-level analysis of recent Demographic and Health SurveysPLOS ONE

Dear Dr. Geberu,

Thank you for submitting your manuscript to PLOS ONE. After careful consideration, we feel that it has merit but does not fully meet PLOS ONE’s publication criteria as it currently stands. Therefore, we invite you to submit a revised version of the manuscript that addresses the points raised during the review process. I have extensibly annotated the original comments and provided feedback about changes to be done in the manuscript in each section. My main concern is about pooling data. First, DHSs data use a two-stage sampling design. When data are pooled I wonder which weights have been used to estimates coefficients (AOR)? Second, the outcome variable is not well operationalized (see manuscript for comments). Are underweight women excluded from analyses. This is unclear from the manuscript. It that is the case, then this introduces additional statistical concern because when estimates are done on sub-populations, additional care should be done to ensure that representativeness of samples are preserved. Third, the discussion section is very flawed since it is like a "redo" of results section. This section should have been more clearer and useful if the authors engaged discussing factors associated with overweight/obesity at (i) individual level; and (ii) community level. The authors should really engage in key findings and how they align or differ from what is already known on the subject; Fourth, authors are referring to high maternal mortality countries without any references on the adopted classification. Fifth, the manuscript requires serious edits.    Please submit your revised manuscript by Aug 21 2024 11:59PM. If you will need more time than this to complete your revisions, please reply to this message or contact the journal office at plosone@plos.org . Please include the following items when submitting your revised manuscript:

We look forward to receiving your revised manuscript.

Kind regards,

Zacharie Tsala Dimbuene, Ph.D.

Academic Editor

PLOS ONE

Reviewers' comments:

Reviewer's Responses to Questions

**Comments to the Author**

1. Is the manuscript technically sound, and do the data support the conclusions?

Reviewer #1: Yes

Reviewer #2: Yes

2. Has the statistical analysis been performed appropriately and rigorously? 

Reviewer #1: Yes

Reviewer #2: Yes

3. Have the authors made all data underlying the findings in their manuscript fully available?

Reviewer #1: Yes

Reviewer #2: Yes

4. Is the manuscript presented in an intelligible fashion and written in standard English?

Reviewer #1: Yes

Reviewer #2: Yes

5. Review Comments to the Author

Reviewer #1: The manuscript is good in terms of objective but not an easy

article to comprehend. The writing should be improved. For example in the abstract the authors written "Women aged 25–34 years and 35-49 years, women who had primary education, who had

secondary education and above, not-working women, married, widowed, and divorced women,

poorer, middle, richer, and richest household, female household head, having one child, 2-3

children, 4 and above children, watching television less than once a week and at least once a week,

using internet less than once a week and at least once a week, improved source of drinking water,

urban residents, low community poverty, and high community level media usage." -- it is not a complete sentence.

I attempted to find the country-specific prevalence of obesity but was unsuccessful in the manuscript. It will be helpful to know the country-specific prevalence before pooling the results.

I recommend that the authors incorporate a discussion of their results in the context of previous studies.

Reviewer #2: General comment

If the initially considered countries were low- and middle-income countries, modify the title to include "high maternal mortality in low and middle-income countries." A flowchart illustrating the number of countries initially considered, and how 21 countries with a sample size of 64,076 were reached, will make the sampling process easier to understand. I also suggest that the outcome variable be overweight/obesity and dichotomized as yes/no.

6. PLOS authors have the option to publish the peer review history of their article (what does this mean? ). If published, this will include your full peer review and any attached files.

**Do you want your identity to be public for this peer review?** For information about this choice, including consent withdrawal, please see our Privacy Policy .

Reviewer #1: No

Reviewer #2: No

---

## [Author Response · Author response to Decision Letter 1]

15 Aug 2024

Point-by-point response for editor’s and reviewers' comments

Editor’s and Reviewers’ comments and suggestions Authors' responses

Editor’s comments and suggestions

1. Thank you for submitting your manuscript to PLOS ONE. After careful consideration, we feel that it has merit but does not fully meet PLOS ONE’s publication criteria as it currently stands. Therefore, we invite you to submit a revised version of the manuscript that addresses the points raised during the review process. Dear Editor, thank you for your consideration and suggestion! We revised the manuscript according to your comments and suggestions.

2. I have extensibly annotated the original comments and provided feedback about changes to be done in the manuscript in each section. My main concern is about pooling data. First, DHSs data use a two-stage sampling design. When data are pooled I wonder which weights have been used to estimates coefficients (AOR)? Dear Editor, thank you for your concern and comment. Pooled data analysis is a methodology of analyzing data from different sources which has the advantage of producing more accurate estimation and greater statistical power. Besides, it helps to generalize findings from different populations. Therefore, by considering the above importance, we conducted pooled data analysis to determine the magnitude of overweight/obesity only. However, we used normal (non-pooled) overweight/obesity categorized (Yes/No) data to conduct multilevel multivariable regression analysis. Hence, we used normal (non-pooled) data to identify the associated factors of the dependent variable and their statistical parameters/coefficients/ such as AOR, COR, P-value, and 95% CI.

Dear editor, as we have tried to describe in the above sentences as to our knowledge and understanding we gave more value for pooled data rather than non-pooled data due to its strong nature of analysis and evidence. In this case, we kindly hope you can understand our intention and this description will answer your comments of pooled data analysis which were raised throughout the document.

3. Second, the outcome variable is not well operationalized (see manuscript for comments). Are underweight women excluded from analyses. This is unclear from the manuscript. It that is the case, then this introduces additional statistical concern because when estimates are done on sub-populations, additional care should be done to ensure that representativeness of samples are preserved. Dear editor, thank you for your comment, for the purpose of further clarity, we have revised and merged the “variable definition” and “operational definition” sections and labeled as “variables and measurements”. Besides, your raised concerns are addressed as per your suggestion, please see the clean manuscript on pages 8-10, lines 169 -210.

Regarding the inclusion/exclusion of underweight women: As we know, our analysis should be done based on our title/topic. Due to this reason, we didn’t include underweight (mildly, moderately and severely thin) women both in the numerator and in the denominator because we are focused to study about the magnitude and associated factors of overweight/obesity as mentioned in the topic. The dependent variable was fixed based on our previous practice in related article [1], other previous published literature [2], and WHO classification [3]. Therefore, as to our knowledge and the practices, there will not be a statistical concern because the dependent variable was fixed by strictly following the standard DHS guide and previous articles.

4. Third, the discussion section is very flawed since it is like a "redo" of results section. This section should have been more clearer and useful if the authors engaged discussing factors associated with overweight/obesity at (i) individual level; and (ii) community level. The authors should really engage in key findings and how they align or differ from what is already known on the subject; Dear editor, thank you for your comment and concern, we revised the discussion according to your suggestion by considering the addressable issues, please see the revised section of the discussion.

5. Fourth, authors are referring to high maternal mortality countries without any references on the adopted classification. Dear editor, thank you very much for your constructive comment, we have mentioned the source of this classification and we also included in the variables and measurements section for further understanding. We hope this definition will answer all comments regarding high maternal mortality, please see the revised clean manuscript on page 10, lines 208-210 and on page 8, line 166.

6. Fifth, the manuscript requires serious edits. Dear editor, thank you for your constructive comments, we have edited and revised each section of the manuscript which needs edition.

7. Why should countries be interested in pooled prevalence? Pragmatically and policy-wise, this is less relevant! Dear editor, thank you for your comment! All the included countries share the same source of information (DHS survey) and for this reason this pooled analysis will have pragmatic policy implications. Thus, to get holistic picture about overweight/obesity among reproductive age women in countries with high maternal mortality need to be evaluated empirically for up-to-date intervention by respective country governments and other partners working on maternal health.

8. Clarify! What is high maternal mortality? Thank you for your question, we have clarified it accordingly. Maternal mortality is considered to be high if it is 300-499, very high if it is 500-999, and extremely high if it is equal to and higher than 1000 maternal deaths per 100,000 live births. For the sake of maintaining economy of words and attractiveness in the abstract we have clarified it in the variables and measurements section, please see the updated clean manuscript on page 10, lines 208-210.

9. DHSs don't allow causality; therefore determinant is inappropriate Dear editor, thank you for your suggestion, we have revised the whole manuscript which states about determinants according to your valuable suggestion, please see the updated manuscript.

10. This is fuzzy in an abstract. Not easy to get a sense of the message conveyed in this long sentence Dear editor, thank you for your constructive comment, for the sake of conciseness in the abstract we tried to shorten it by mentioning factors having more strong association. Please see the revised section of the abstract on page 3, lines 65 to 70. Our assumption when we omit these less strong factors is that readers can get the detail of all factors on the result and discussion section.

11. General comment: The English is poor and needs serious improvement! Dear editor, thank you for your comment, we have revised and edited the entire manuscript by language expert to improve the language of the manuscript.

Besides, we revised the background section according to your specific comments and tried to include these specific comments’ response in this point-by-point response.

12. This kind of construction breaks the flow of the paper. Talk first of prevalence from previous studies globally, in LMICs, and at country-level. Thereafter, move to consequences, and associated factors. Dear editor, thank you for your insights and improvement suggestion, we revised according to your suggestion, please see the revised clean manuscript on page 5-6, lines 94-120.

13. It is not watching TV per se. It is about "sedentary behaviour" Dear editor, you are correct, media usage in general is related with sedentary behavior which in return affects overweight and we have clearly stated on the possible justification of the discussion section. However, in the background section, for the purpose of specificity we have presented as frequency of watching television.

14. what's the added value of pooled prevalence Dear editor, thank you for your clarification question! Pooled data analysis is a methodology of analyzing data from different sources which has the advantage of producing more accurate estimation and greater statistical power. Besides, it helps to generalize findings from different populations. Therefore, by considering the above importance, we conducted pooled data analysis to determine the magnitude.

15. Readers were expecting till now to know (understand) how high maternal mortality was defined. Nothing but speculations Dear editor, thank you for your comment again, we defined in the “variables and measurement” section.

16. When data are pooled, how did the authors incorporate the complex survey design (CSD) in DHSs? Dear editor, thank you for your question, as mentioned in the method part of the manuscript, cross-sectional study design was used in all Demographic and Health Surveys. Therefore, there will no be a problem of complex survey design.

17. This is what? It is well known that DHSs collect information on women of reproductive ages (15-49 years). What this sentence adds up? Dear editor, yes you are correct it didn’t have value and thank you for your valuable comment, we removed it.

18. Data sources for these statistics ? Dear editor, thank you for your question, we mentioned the data source, please see at the bottom of table 1. We have used three data sources as follows: http://www.dhsprogram.com, http://srhr.org/mmr2020, or http://www.who.int/publications/i/item/9789240068759

19. Whose classification is it? Dear editor, thank you for your clarification question, this is the classification of WHO, UNICEF, and world bank. We have cited the appropriate reference, please see the revised clean manuscript on page 8, line 166.

20. This is vague while this key for the manuscript. How was it collected in DHSs? How was it treated in this paper? How the decision to operationalize the original variable was taken ? Dear editor, thank you for your valuable comments and insights, we revised it according to your comment, please see the “variable and measurement” section on pages 8-9, lines 171-185.

21. This should go to "outcome variable?" Dear editor, thank you for your valuable suggestion, we revised it accordingly, please see on pages 8-9, lines 176-179.

22. why capitalize? Thank you for your question! We had a practice of capitalizing the first spelling of abbreviated phrases and that is why we capitalized the first spellings.

23. This is debatable since even non-bivariate associations could become "significant" in multivariable analyses. this choice should be justified Dear editor, thank you for your comment! Yes, it is debatable, however, as the practice of different scholars we used 0.25 as a criterion to filter our candidate variables for multilevel regression analysis.

24. This section is too log for secondary data analysis Dear editor, yes you are correct, it is long. However, we did this to answer the questions which were raised by the PLOS ONE journal assistants during technical check as journal requirement.

25. For binary variables and to make Table clearer, knowing "one category" reveals info on the other category. Dear editor, thank you for your comment and you are correct.

26. I wonder whether these 2 sub-headings are necessary (Random vs. Fixed effects) Dear editor, thank you for your suggestion, we corrected is as per your suggestion, please see the revised manuscript on page 15.

27. This is not the numbers of women of reproductive ages in DHS report? How do we trust these numbers ? Dear editor, thank you for your clarification question, this is the number of non-pregnant, who didn’t have births in the last five years, and non-underweight women which was fixed according to the DHS statistics guide. Additionally, we reached to this number after omitting all missing variables, so it might not be equal to the observed reproductive age women. For further understanding please see the flow chart named as figure 1 and attached as TIF file, please see the uploaded figure.

28. It should have been a good idea to split discussion into 2 sub-headings: Individual-level factors and community-level factors and better engage the discussion and provide plausible explanations to understand findings Dear editor, we understand your concerns on the discussion and appreciate your specific comments in the manuscript. However, we face difficulties to split the discussion in to two sub-headings. Because if we split in to two sub-headings there will be redundancy of possible justifications between some individual level and community level factors which might affect the attractiveness of this section. For example; frequency of watching television from individual level factors and community-level media usage from community level factor had association with the dependent variable. In these comparable variables we had similar possible justifications. If we split these comparable variables in to different sections and if we state the similar possible justification, repetition of idea might occur.

29. This is a less productive way to engage in a discussion, which shouldn't be a "redo" of results Dear editor, thank you for your comment, we presented the factors that need modification in synthesized manner, please see on page 19, lines 299-305.

30. what's functional performance? Dear editor, thank you for your clarification point, we revised the full sentence to render full meaning, please see the clean manuscript on page 19, lines 310-311.

31. This should be dug enough Dear editor, thank you for your critical insight, we revised it in detail, please see the clean manuscript on page 20, lines 319-330.

Reviewer 1: comments and suggestions

1. The manuscript is good in terms of objective but not an easy

article to comprehend. The writing should be improved. For example in the abstract the authors written "Women aged 25–34 years and 35-49 years, women who had primary education, who had

secondary education and above, not-working women, married, widowed, and divorced women,

poorer, middle, richer, and richest household, female household head, having one child, 2-3

children, 4 and above children, watching television less than once a week and at least once a week,

using internet less than once a week and at least once a week, improved source of drinking water,

urban residents, low community poverty, and high community level media usage." -- it is not a complete sentence. Dear reviewer, thank you very much for your valuable comment, we modified this sentence, please see the abstract on page 3, lines 65-70.

2. I attempted to find the country-specific prevalence of obesity but was unsuccessful in the manuscript. It will be helpful to know the country-specific prevalence before pooling the results. Dear reviewer, thank you for your comment, the country specific prevalence was done and stated on figure 2 which was uploaded as TIF file separately, please see the uploaded figure.

3. I recommend that the authors incorporate a discussion of their results in the context of previous studies. Dear reviewer, thank you in advance for your recommendation, we have discussed the magnitude from our result and the associated factors from previous studies. We have discussed the magnitude from our result purposely because our study is about pooled prevalence and we didn’t found pooled prevalence of overweight/obesity form previous studies. In this case if we compare our pooled prevalence from previous single prevalence studies, this will lead us to compare incomparable magnitudes/findings. This is the reason why we had discussed the magnitude by using our result only. However, we have discussed the associated factors from previous studies in detail which is inline with your valuable recommendation.

Reviewer 2: comments and suggestions

1. 1 If the initially considered countries were low- and middle-income countries, modify the title to include "high maternal mortality in low and middle-income countries." Dear reviewer, thank you very much for your insight, we revised the title according to your constructive comment. Besides, the rest of the manuscript is also modified according to the slight modification in the title.

2. 2 A flowchart illustrating the number of countries initially considered, and how 21 countries with a sample size of 64,076 were reached, will make

---

## [Decision Letter · Decision Letter 1]

14 Oct 2024

PONE-D-23-42732R1Pooled prevalence and factors of overweight/obesity among reproductive age women in low and middle-income countries with high maternal mortality:  A multi-level analysis of recent Demographic and Health SurveysPLOS ONE

Dear Dr. Geberu,

Thank you for submitting your manuscript to PLOS ONE. After careful consideration, we feel that it has merit but does not fully meet PLOS ONE’s publication criteria as it currently stands. Therefore, we invite you to submit a revised version of the manuscript that addresses the points raised during the review process.

1) The study excluded, unjustifiably, from the study underweighted women. Similar studies (see reference provided in annotated manuscript, did not excluded underweigthed women. Authors based this irrationale decision on studies conducted in Ethiopia. This needs to be addressed.  2) In either case, analyses are based on sub-samples in each country because not all women have information on BMI. Therefore, using svyset, and svy are not enough to account for complex sampling design. This is also need to addressed correctly. 3) Many variables at community-level were collected at woman-level. Authors need to clarify how these variables were aggregated at community-level. 4) The paper lacks bivariate analyses. From Table 2 (descriptives of the sample), the paper jumps to multi-level analyses.  Thank you

We look forward to receiving your revised manuscript.

Kind regards,

Zacharie Tsala Dimbuene, Ph.D.

Academic Editor

PLOS ONE

Reviewers' comments:

Reviewer's Responses to Questions

**Comments to the Author**

1. If the authors have adequately addressed your comments raised in a previous round of review and you feel that this manuscript is now acceptable for publication, you may indicate that here to bypass the “Comments to the Author” section, enter your conflict of interest statement in the “Confidential to Editor” section, and submit your "Accept" recommendation.

Reviewer #2: All comments have been addressed

Reviewer #3: All comments have been addressed

2. Is the manuscript technically sound, and do the data support the conclusions?

Reviewer #2: Yes

Reviewer #3: Yes

3. Has the statistical analysis been performed appropriately and rigorously? 

Reviewer #2: Yes

Reviewer #3: Yes

4. Have the authors made all data underlying the findings in their manuscript fully available?

Reviewer #2: Yes

Reviewer #3: Yes

5. Is the manuscript presented in an intelligible fashion and written in standard English?

Reviewer #2: Yes

Reviewer #3: Yes

6. Review Comments to the Author

Reviewer #2: The authors have addressed all of my comments and I accepted the manuscript for publication in its current form.

Reviewer #3: (No Response)

7. PLOS authors have the option to publish the peer review history of their article (what does this mean? ). If published, this will include your full peer review and any attached files.

**Do you want your identity to be public for this peer review?** For information about this choice, including consent withdrawal, please see our Privacy Policy .

Reviewer #2: No

Reviewer #3: No

---

## [Author Response · Author response to Decision Letter 2]

19 Oct 2024

Point-by-point response for editor’s and reviewers' comments

Editor’s and Reviewers’ comments and suggestions Authors' responses

Editor’s comments and suggestions

1. Thank you for submitting your manuscript to PLOS ONE. After careful consideration, we feel that it has merit but does not fully meet PLOS ONE’s publication criteria as it currently stands. Therefore, we invite you to submit a revised version of the manuscript that addresses the points raised during the review process. Dear Editor, We would like to extend our sincere thanks to you and the reviewers for your invaluable feedback and critical insights during the two rounds of revisions. Your thoughtful comments have greatly contributed to the improvement of our manuscript, helping to enhance its quality and clarity. We deeply appreciate the time and effort you and the reviewers invested in providing detailed suggestions, which have led to significant improvements in the content and overall structure of the paper. We have carefully addressed all the feedback and revised the manuscript accordingly. Thank you once again for your support and guidance.

2. This means what? Should you really expect same number of children for all women?

Dear Editor, thank you for your clarification request. We did not intend to suggest that all women would have the same number of children. The use of the word "different" was meant to highlight the variation across categories. However, to avoid any confusion for readers, we have revised the sentence accordingly. Please refer to the revised section of the abstract in the clean version of the manuscript, located on page 3, line 69.

3. This is not TRUE since the study is done, injustifiably, on a sub-sample of women (normal, overweight/obese)

Dear editor, thank you for your concerns, in the current study, we recalculated the sampling weights specifically for the sub-sample of women with valid Body Mass Index (BMI) data, excluding those categorized as underweight. The original DHS weights were designed to apply to the full survey population, but because not all women had BMI data, and our focus was on women with normal, overweight, or obese BMI levels, recalculating the weights ensured that our sub-sample remained representative of the population of interest. This approach preserves the integrity of the DHS complex survey design by maintaining accurate weighting adjustments for the primary sampling units (PSUs) and strata. Please kindly see the updated version of the manuscript on pages 7-8, lines 162-166.

4. 1) The study excluded, unjustifiably, from the study underweighted women. Similar studies (see reference provided in annotated manuscript, did not excluded underweigthed women. Authors based this irrationale decision on studies conducted in Ethiopia. This needs to be addressed. Dear Editor, thank you in advance for your thoughtful comments and feedback. Genuinely, all of your comments and suggestions are really invaluable for our manuscript significant improvement and we are lucky to get such vibrant research scientist.

We appreciate your concern regarding the exclusion of underweight women from the study. This decision was made with the aim of focusing specifically on the distinct public health challenges associated with overweight and normal-weight women. We acknowledge that underweight women face their own unique health risks, but including them in our analysis could have diluted our findings and shifted the focus away from the primary aim of addressing the rising burden of overweight and obesity (which is one of the current neglected public health problem specially in developing countries) in the population under study.

Although some studies have included underweight women, several authors have similarly chosen to exclude this group when concentrating on overweight and obesity to allow for a clearer understanding of the risk factors and health outcomes associated with these conditions. Our decision is in line with this practice, as we wanted to provide targeted insights on these critical public health concerns without compromising the specificity of the results.

Nevertheless, we recognize this exclusion as a limitation of our study and have clearly acknowledged this in the manuscript, please kindly see on page 27, lines 420-421. We understand that future research should explore the specific challenges of underweight women, but our current focus was on the health risks of overweight and obesity.

5. 2) In either case, analyses are based on sub-samples in each country because not all women have information on BMI. Therefore, using svyset, and svy are not enough to account for complex sampling design. This is also need to addressed correctly. Dear editor, thank you for your concerns, we would like to confirm that the issue of sub sampling and complex sampling design has been already addressed.

The DHS weights are originally designed for the full sample. However, since not all women had BMI data, we recalculated the sampling weights specifically for the sub-sample of women with valid BMI data. This adjustment ensures that the sub-sample remains representative of the population in each country while maintaining the integrity of the survey’s complex design.

We understand your concern that simply applying the svyset and svy commands may not fully account for complex sampling. To address this, we updated the svyset and svy commands with the recalculated weights to correctly reflect the primary sampling units (PSUs) and strata for the sub-sample. This process ensures that the complex sampling design is properly accounted for, even when working with a sub-sample.

We revised the manuscript according to the above explanation, please kindly see the updated version of the manuscript on pages 7-8, lines 162-166.

6. 3) Many variables at community-level were collected at woman-level. Authors need to clarify how these variables were aggregated at community-level.

All these variables were collected at woman-level. Clearly explicit how aggregation was done. Dear editor, thank you very much for your valuable suggestions. We have addressed your concerns in the updated version of the manuscript. Please kindly see the updated manuscript on pages 10-11, lines 197-211.

7. 4) The paper lacks bivariate analyses. From Table 2 (descriptives of the sample), the paper jumps to multi-level analyses.

This Table is not well designed. Use categories in same column with variables

Additionally, this Table should include the dependent variable ?

Indeed, Table 3 jumps into modeling without any indication of bivariate analyses ???? Dear editor, thank you in advance for your invaluable suggestions and comments. We have addressed all of your concerns in the updated version of the manuscript. Please kindly see the updated manuscript on pages 15-18, lines 265-270.

8. Why so many variables on medias? You should have created an index of media exposure as most studies do?

Dear Editor, thank you very much for your suggestion and guidance. However, we put media variables in separate manner purposely unlike previous studies. The reason of putting them in separate format is to clearly delineate their specific individual effect on the outcome of interest. We believe analyzing such specific association might have an advantage of rendering specific recommendation for each media component.

9. Why do you need this elliptic format? Dear editor, thank you for your question, we used this elliptic format to visibly differentiate form the rest of the boxes. However, if it has an effect of confusing readers, we adjusted it in to rectangle, please kindly see the revised flowchart.

10. The direction of the arrow does not make sense

Exclude should go out not IN Dear editor, thank you for your valuable suggestion, we adjusted it according to your suggestion, please kindly see the revised flowchart.

11. This last number is unnecessary and misleading Dear editor, thank you for your comment, however, since this last number is the weighted sample which was used for each of our analysis, omitting this value might confuse the readers because the readers might use the unweighted frequency incorrectly. As a result, as to our understanding, mentioning in the flowchart might be advantageous.

---

## [Decision Letter · Decision Letter 2]

14 Jan 2025

PONE-D-23-42732R2Pooled prevalence and factors of overweight/obesity among women of reproductive age in low and middle-income countries with high maternal mortality:  A multi-level analysis of recent Demographic and Health SurveysPLOS ONE

Dear Dr. Geberu,

Thank you for submitting your manuscript to PLOS ONE. After careful consideration, we feel that it has merit but does not fully meet PLOS ONE’s publication criteria as it currently stands. Therefore, we invite you to submit a revised version of the manuscript that addresses the points raised during the review process.

**The manuscript has been evaluated by one reviewer, and their comments are available below.**

**The reviewer has raised a number of major concerns, particularly around the methods section and statistical approach, as well as the presentation of the results.**

Could you please carefully revise the manuscript to address all comments raised?

We look forward to receiving your revised manuscript.

Kind regards,

Johanna Pruller, Ph.D.

Associate Editor

PLOS ONE

Reviewers' comments:

Reviewer's Responses to Questions

**Comments to the Author**

1. If the authors have adequately addressed your comments raised in a previous round of review and you feel that this manuscript is now acceptable for publication, you may indicate that here to bypass the “Comments to the Author” section, enter your conflict of interest statement in the “Confidential to Editor” section, and submit your "Accept" recommendation.

Reviewer #2: All comments have been addressed

Reviewer #3: All comments have been addressed

Reviewer #4: (No Response)

2. Is the manuscript technically sound, and do the data support the conclusions?

Reviewer #2: Yes

Reviewer #3: Yes

Reviewer #4: Partly

3. Has the statistical analysis been performed appropriately and rigorously? 

Reviewer #2: Yes

Reviewer #3: Yes

Reviewer #4: No

4. Have the authors made all data underlying the findings in their manuscript fully available?

Reviewer #2: Yes

Reviewer #3: Yes

Reviewer #4: No

5. Is the manuscript presented in an intelligible fashion and written in standard English?

Reviewer #2: Yes

Reviewer #3: Yes

Reviewer #4: Yes

6. Review Comments to the Author

**Reviewer #2: ** The authors addressed the comments and concerns I raised in revision one. I have no further comments.

**Reviewer #3:**  (No Response)

**Reviewer #4: ** I come to this paper as a new reviewer on this revision and struggle with the way the findings are worded here. The associations are between variables and not between given categories (abstract); the issue here of the risk factors being potentially associated- use of internet may imply a higher socioeconomic grouping - needs to be dealt with in the summary of the results. The idea that the highest socioeconomic group needs more targeting is against the evidence showing an increasing issue in the lower groups - as this may the group where the targeting may have best effect.

Please do not use "includes" for countries but list all countries in the study. There is a preponderance of African countries here - please explain the cut-offs used that cause a concentration on these countries and why other countries do not qualify.

Above community is country and issues such as conflict, natural disaster etc which are known to impact maternal morbidity and mortality - were these taken into account?

One may argue there is reverse causation in things like marital status - how was this dealt with here?

Age and number of living children is clearly correlated - how do you unpick this?

In what sense can things be explained by older age and lack of poverty - both of which are known to be associated with higher BMI?

7. PLOS authors have the option to publish the peer review history of their article (what does this mean? ). If published, this will include your full peer review and any attached files.

**Do you want your identity to be public for this peer review?** For information about this choice, including consent withdrawal, please see our Privacy Policy .

Reviewer #2: No

Reviewer #3: No

Reviewer #4: No

---

## [Author Response · Author response to Decision Letter 3]

4 Feb 2025

Point-by-point response for editor’s and reviewers' comments

Sr.No Editor’s and Reviewers’ comments and suggestions Authors' responses

Editor’s comments and suggestions

1. The reviewer has raised a number of major concerns, particularly around the methods section and statistical approach, as well as the presentation of the results.

Could you please carefully revise the manuscript to address all comments raised? Dear Editor, We would like to extend our sincere thanks to you and the reviewers for your invaluable feedback and critical insights during this the third rounds of revisions. Since we got comments only from reviewer 4 (the other reviewers didn’t have comments at this stage of revision process), we have carefully addressed the feedback and revised the manuscript accordingly.

Reviewer four’s comments and suggestions

1 I come to this paper as a new reviewer on this revision and struggle with the way the findings are worded here. The associations are between variables and not between given categories (abstract); the issue here of the risk factors being potentially associated- use of internet may imply a higher socioeconomic grouping - needs to be dealt with in the summary of the results. Dear reviewer, thank you in advance for your valuable suggestions. Yes, you are write, the association is between the dependent and independent variables and we have adjusted the confusing sentence in order not to confuse other further readers. Please kindly see the updated manuscript at the abstract section on page 3, line 67.

Regarding the relationship between use of internet and higher socioeconomic grouping, thank you again for your insight and we have addressed your comment at the discussion section (in order not to congest the summary of results, we prefer to incorporate at the discussion part). Please kindly see the updated manuscript at the discussion section on page 22, lines 373-376.

2 The idea that the highest socioeconomic group needs more targeting is against the evidence showing an increasing issue in the lower groups - as this may the group where the targeting may have best effect. Dear reviewer, thank you for your comment. If we get your idea, we have said that socio-economic status and weight gain has direct relationship i.e the richest are developing more weight gain than the poorest in developing countries. Therefore, we kindly request you to understand our description as we didn’t say the lower socio-economic group are related with an increasing issue. Probably, if we made confusion by the phrase “low community poverty”, having low community poverty indicates higher economic status than high community poverty. For further clarification, please kindly see the revised “variables and measurement” section on page 10, lines 216-218.

Moreover, the previous articles suggested that socio-economic status has different effect on weight gain of developing and developed countries and for further clarification, we have modified the discussion section please kindly see on page 23, lines 386-390..

However, since our study is focusing on developing countries, our major analysis and interpretation is mainly targeting to these developing countries.

3 Please do not use "includes" for countries but list all countries in the study. Dear reviewer, thank you very much for your suggestion. We modified the sentence according to your suggestion. Please kindly see the updated manuscript at the method section on page 7, lines 145-146.

4 There is a preponderance of African countries here - please explain the cut-offs used that cause a concentration on these countries and why other countries do not qualify. Dear reviewer, thank you for your clarification question. Our study is conducted by focusing only on countries with high maternal mortality. We used WHO cut-off value to classify maternal mortality as follows: The maternal mortality ratio (MMR) is categorized as high if it falls within the range of 300-499, very high if it is between 500-999, and extremely high if it equals or exceeds 1000 maternal deaths per 100,000 live births. This classification sentence was stated at the “variables and measurement” section, please kindly see it on page 10, lines 223-225. Besides, this cut-off value was mentioned under table 1, please kindly see on page 8, line169.

The reason of focusing on those selected countries was due to the women in such countries seeks an urgent attention to avert causes of morbidity and mortality. Therefore, other countries that have maternal mortalities other than high, very high, and extremely high did not qualify for this study and were excluded. That is why our studied countries were concentrated majorly on Africa.

5 Above community is country and issues such as conflict, natural disaster etc which are known to impact maternal morbidity and mortality - were these taken into account? Dear reviewer, thank you for your critical insight. We were planned to conduct sub-group analysis by region in order to address your specific comments, however, by default majority of the countries fall under a single sub-group (sub-Saharan African countries), which was difficult to undertake sub-group analysis since the category didn’t fulfil chi-square assumption. Therefore, we can’t consider country specific variables like conflict and natural disaster.

6 One may argue there is reverse causation in things like marital status - how was this dealt with here? Dear reviewer, thank you for your comment. Your critical comment improved our manuscript in advance. Yes! You are write, even if majority of the findings support our research result, there could be a reverse causation and we modified our discussion section according to your valuable comment, please kindly see on page 23, lines 406-408.

7 Age and number of living children is clearly correlated - how do you unpick this? Dear reviewer, thank you for your comment and we want to share your idea by considering ourselves as a reviewer. However, as a researcher, we were expected to conduct multicollinearity assessment to identify the relationship between independent variables. Accordingly, the result depicted that there were no issues found as all variables had VIF <5, with the best fitted model (model III) VIF at 1.74, please kindly see on page 11, lines 240-241. Therefore, based on the finding of the above statistical parameter, we analyzed age and number of living children as an independent factors in our final output table.

8 In what sense can things be explained by older age and lack of poverty - both of which are known to be associated with higher BMI? Dear reviewer, thank you for your clarification question. Our study includes only the reproductive age group women (15-49 years). The positive association between older age and weight gain might be attributed to physiological changes that occur with age, such as hormonal shifts, increased body fat, and changes in body composition. Additionally, reduced participation in physical activities due to age-related physiological changes among women may contribute to the increased weight gain. Moreover, it might be related with our basal metabolic rate, which is the rate at which our body utilizes energy during at rest to keep vital body functions going, that slow down year to year which resulted in high caloric density and weight gain later on. Furthermore, the shifts in behavior, lifestyle and even emotional health might be related with weight gain in older age. For further clarification, we added more additional possible justifications, please kindly see at the discussion section on page 21, lines 331-335.

When we come to the relationship between lack of poverty and weight gain, you raised an important point and we addressed your concern as follows:

Lack of poverty (high socio-economic status) has different association with weight gain in developing and developed countries. In developing countries, it has direct relationship with weight gain i.e; individuals with higher wealth are more inclined to consume energy-dense foods and lead sedentary lifestyles. Additionally, women in higher wealth quantiles might have a higher caloric intake compared to those in lower quantiles, which could be a primary factor contributing to weight gain.

However, paradoxically in developed countries, there was an inverse association between economic status and weight gain which depicted that lower economic status was related with higher weight gain. The suggested possible justification could be that poor socio-economic groups in developed countries are more likely to be unemployed, lower educated and might have irregular meals. Therefore, according to your constructive suggestion, we revised the discussion section please take a look on page 23, lines 386-390.

---

## [Decision Letter · Decision Letter 3]

15 May 2025

PONE-D-23-42732R3Pooled prevalence and factors of overweight/obesity among women of reproductive age in low and middle-income countries with high maternal mortality:  A multi-level analysis of recent Demographic and Health SurveysPLOS ONE

Dear Dr. Geberu,

Thank you for submitting your manuscript to PLOS ONE. After careful consideration, we feel that it has merit but does not fully meet PLOS ONE’s publication criteria as it currently stands. Therefore, we invite you to submit a revised version of the manuscript that addresses the points raised during the review process.

Three reviewers have assessed your revised submission with two requesting additional revisions before your manuscript can be considered for publication. Please carefully respond to all comments, including the nuance and challenges with BMI as an indicator of weight and health.  Please submit your revised manuscript by Jun 29 2025 11:59PM. If you will need more time than this to complete your revisions, please reply to this message or contact the journal office at plosone@plos.org . Please include the following items when submitting your revised manuscript:

We look forward to receiving your revised manuscript.

Kind regards,

Jennifer Tucker, PhD

Staff Editor

PLOS ONE

Reviewers' comments:

Reviewer's Responses to Questions

**Comments to the Author**

1. If the authors have adequately addressed your comments raised in a previous round of review and you feel that this manuscript is now acceptable for publication, you may indicate that here to bypass the “Comments to the Author” section, enter your conflict of interest statement in the “Confidential to Editor” section, and submit your "Accept" recommendation.

Reviewer #3: All comments have been addressed

Reviewer #4: (No Response)

Reviewer #5: (No Response)

2. Is the manuscript technically sound, and do the data support the conclusions?

Reviewer #3: Yes

Reviewer #4: (No Response)

Reviewer #5: Yes

3. Has the statistical analysis been performed appropriately and rigorously? 

Reviewer #3: Yes

Reviewer #4: (No Response)

Reviewer #5: Yes

4. Have the authors made all data underlying the findings in their manuscript fully available?

Reviewer #3: Yes

Reviewer #4: (No Response)

Reviewer #5: Yes

5. Is the manuscript presented in an intelligible fashion and written in standard English?

Reviewer #3: Yes

Reviewer #4: Yes

Reviewer #5: (No Response)

6. Review Comments to the Author

Reviewer #3: (No Response)

Reviewer #4: Thank you for your response to my question.

I'm afraid the issue of categories versus variables remains. The offending line in the abstract is not just line 67 and it is not the issue of the word assocaited. The factors associated are age,marital status, household income, household size etc. Please talk about the variables throughout the paper e.g. al;so the paragraph starting line 293. Connected with this Table 4 should not bold and use stars for significant categories but do an overall test for association at a variable level (see for example the seminal work of Tukey on this for an explanation of why this is not statistically valid).

The issue of potentially associated variables needs to be part of the analyses - we know the issue with amyl nitrite and AIDS, and we need to understand whether variable selection plays a part - the issue of what is potentially a confounding factor is a result not a discussion point.

The issue in developed countries and developing countries is the different types of diet that are associated with poverty - this is well known.

While here the higher socioeconomic groups have higer rates of obesity; it would help to understand if there are trends of greater increases over time in lower sciopcioeconomic groups which might indicate that as these are developing bad habits but might be persuaded to change, this is an important group to target.

It is entirely possible to site this inside a further level in a multilevel model to allow for things that affect people at a country level.

It is known that BMI tends to increase with age even in premenopausal women - this needs to be unpicked as above.

Reviewer #5: The manuscript, titled "Pooled prevalence and factors of overweight/obesity among women of reproductive age in low and middle-income countries with high maternal mortality: A multi-level analysis of recent Demographic and Health Surveys," presents an important topic. The study effectively highlights the prevalence and determinants of overweight/obesity among women in LMICs with high maternal mortality. The statistical approach is well-defined, and the multilevel analysis is appropriate. However, a few concerns remain, particularly regarding clarity in result presentation and language consistency. As I am a new reviewer to this manuscript, in addition to reviewing the manuscript, I have also examined whether the issues raised by the previous reviewer have been addressed or not. I appreciate that the authors have taken the comments of previous reviewer into consideration and made changes in the manuscript. Here is the detailed review of the manuscript:

• Line 67: The authors have revised the phrasing of associations to indicate relationships between variables instead of categories, addressing the previous reviewer’s concern. However, minor rewording for clarity would enhance readability.

• Line 73-76: The recommendation section mentions targeting "older women, those with formal education, non-working women, individuals who spend time watching television and using the internet, urban residents, female household heads, and women in the wealthiest households."

o This contradicts the previous reviewer's concern that the highest socioeconomic group may not be the most urgent target (previous reviewer's comment). The phrasing should be revised to reflect a more balanced intervention approach.

• Line 92-93: The statement about overweight/obesity increasing among lower economic groups over time should be further clarified. Does this apply to all LMICs or only a subset?

• Line 131-137: The justification for the study’s objective is clear, but it would help to specify how this study improves on existing literature in terms of methodology or scope.

• Line 145-146: The sentence listing countries has been revised, addressing the previous reviewer’s concern about the use of “includes.”

• Line 223-225: The classification of maternal mortality cut-offs using WHO guidelines has been included. This addresses the previous reviewer’s request to explain why African countries dominate the dataset. However, a brief mention of why high maternal mortality is particularly relevant to obesity research would strengthen the rationale.

• Line 240-241: The authors checked multicollinearity (VIF < 5) to ensure that age and number of living children could be analyzed independently. This adequately addresses the previous reviewer’s concern. However, it may be beneficial to explain how interaction effects were considered, if applicable.

Line 272-275: The pooled prevalence of overweight/obesity is well reported. However, it might be useful to include a brief mention of which factors contribute to variations between countries to strengthen the analysis.

• Line 279-282: The discussion of ICC and MOR in the null model is useful. However, the interpretation of ICC (19%) could be expanded to explain whether this level suggests a strong or moderate clustering effect.

• Line 296-300: The separation of individual and community-level factors is well done. However, the inclusion of community poverty as a determinant still does not fully address the previous reviewer’s concern about targeting higher-income groups. The discussion should acknowledge this limitation.

• Line 373-376: The authors have moved the discussion of internet use as a proxy for higher socioeconomic status to the discussion section instead of the summary of results. This revision is appropriate. The authors could further acknowledge that internet use itself is not a direct cause of obesity, but rather a proxy for wealth and lifestyle behaviours.

• Line 386-390: The revision addressing the difference between obesity patterns in developing vs. developed countries is well-written. However, the phrase "low community poverty" is still potentially misleading. Consider replacing it with "higher economic status at the community level."

• Line 406-408: The authors acknowledge the possibility of reverse causation regarding marital status and obesity and provide a brief discussion on this. This improves the credibility of their analysis.

• Statistical Models and Interpretation (Table 4, Multilevel Logistic Regression Analysis, Page 30-31): The authors provided a detailed breakdown of their models, but further discussion on effect sizes and practical implications of findings could enhance clarity.

Recommendations:

• The discussion still suggests that the wealthiest groups should be the focus of intervention, contradicting trends in lower-income groups. Further justification with stronger evidence would help.

• The term "low community poverty" is misleading and should be replaced.

• ICC interpretation should be more explicit about its strength.

With these minor refinements, the manuscript would be significantly improved. The methodological rigor and data analysis remain sound, making the study a valuable contribution to public health research in LMICs.

Overall, the authors have made most of the necessary changes, but a final round of refinements would improve readability and interpretation.

7. PLOS authors have the option to publish the peer review history of their article (what does this mean? ). If published, this will include your full peer review and any attached files.

**Do you want your identity to be public for this peer review?** For information about this choice, including consent withdrawal, please see our Privacy Policy .

Reviewer #3: No

Reviewer #4: No

Reviewer #5: No

---

## [Author Response · Author response to Decision Letter 4]

17 Jun 2025

Point-by-point response for editor’s and reviewers' comments

Sr.No Editor’s and Reviewers’ comments and suggestions Authors' responses

Editor’s comments and suggestions

1. Thank you for submitting your manuscript to PLOS ONE. After careful consideration, we feel that it has merit but does not fully meet PLOS ONE’s publication criteria as it currently stands. Therefore, we invite you to submit a revised version of the manuscript that addresses the points raised during the review process.

Three reviewers have assessed your revised submission with two requesting additional revisions before your manuscript can be considered for publication. Please carefully respond to all comments, including the nuance and challenges with BMI as an indicator of weight and health.

Dear Editor, thank you for your valuable support in helping to improve the quality of our manuscript. We have carefully considered the reviewers’ comments and invested significant time and research effort to address each point thoroughly. The manuscript has been revised accordingly.

Regarding the nuances and challenges:

Although BMI has certain nuances and limitations as an indicator of weight and health, we used it as a measurement tool for the following reasons:

1. Body Mass Index (BMI) is a widely accepted metric for assessing body weight in relation to height and is commonly used as a standard indicator of overall health;

2. We followed the standard guidelines provided in the DHS (Demographic and Health Surveys) to statistics, where BMI is used as a key measure of the nutritional status of both women and men;

3. Although waist circumference, waist-to-hip-ratio, and body composition analysis are considered as more accurate measure, they are not available in the DHS dataset. Therefore, BMI remains the most appropriate and accessible measure within the DHS data;

4. BMI is widely relied upon in public health and clinical guidelines as a primary indicator for assessing weight-related health risks.

Therefore, the above are sound reasons to use BMI as a measurement in the DHS data set.

However, to notify the more appropriate measurement for future researchers, we included the nuances and challenges as a limitation, please kindly see the limitation section on pages 27-28, lines 459-465.

Reviewer four’s comments and suggestions

1. I'm afraid the issue of categories versus variables remains. The offending line in the abstract is not just line 67 and it is not the issue of the word assocaited. The factors associated are age,marital status, household income, household size etc. Please talk about the variables throughout the paper e.g. al;so the paragraph starting line 293. Connected with this Table 4 should not bold and use stars for significant categories but do an overall test for association at a variable level (see for example the seminal work of Tukey on this for an explanation of why this is not statistically valid). Dear reviewer, thank you very much for clarifying your previous comment and for giving us this additional opportunity. We initially mentioned each category of variables showing statistically significant associations to provide clearer specification and better understanding for readers. However, in response to your valid and constructive feedback, we have revised the entire document accordingly, please kindly see on the revised manuscript on page 3, lines 67-71 and pages 17-18, lines 303-309.

Regarding the bolding and stars in the categories of Table 4, we sincerely appreciate your insightful comments. We have revised the entire table based on your valuable suggestions and have applied John Tukey’s method accordingly, please kindly see table 4 on page 19-22.

2. The issue of potentially associated variables needs to be part of the analyses - we know the issue with amyl nitrite and AIDS, and we need to understand whether variable selection plays a part - the issue of what is potentially a confounding factor is a result not a discussion point. Dear reviewer, thank you for your constructive comment. As our study relied on a secondary data source, the specific variables you mentioned were not available in the dataset. This limitation is inherent to the use of secondary data, and as a result, we were unable to include these important variables in our analysis.

3. The issue in developed countries and developing countries is the different types of diet that are associated with poverty - this is well known.

While here the higher socioeconomic groups have higer rates of obesity; it would help to understand if there are trends of greater increases over time in lower sciopcioeconomic groups which might indicate that as these are developing bad habits but might be persuaded to change, this is an important group to target. Dear reviewer, thank you for your kind understanding.

Yes, the difference between developed and developing countries lies in the types of diets commonly consumed by these populations.

The odds ratios are a clear indicative of the increment of obesity i.e; 1.3 (poorer), 1.75 (middle), 1.98 (richer), and 2.38 (richest). That is why we recommend special attention for higher socio-economic group. Regarding these, we have done a slight modification in the discussion, please kindly see on page 25, lines 392-395.

4. It is known that BMI tends to increase with age even in premenopausal women - this needs to be unpicked as above. Dear reviewer, thank you for your valuable and critical insight. We have revised the discussion section in accordance with your constructive comment, please kindly see on pages 23-24, lines 336-344.

Reviewer five’s comments and suggestions

1. The manuscript, titled "Pooled prevalence and factors of overweight/obesity among women of reproductive age in low and middle-income countries with high maternal mortality: A multi-level analysis of recent Demographic and Health Surveys," presents an important topic. The study effectively highlights the prevalence and determinants of overweight/obesity among women in LMICs with high maternal mortality. The statistical approach is well-defined, and the multilevel analysis is appropriate. However, a few concerns remain, particularly regarding clarity in result presentation and language consistency. As I am a new reviewer to this manuscript, in addition to reviewing the manuscript, I have also examined whether the issues raised by the previous reviewer have been addressed or not. I appreciate that the authors have taken the comments of previous reviewer into consideration and made changes in the manuscript. Here is the detailed review of the manuscript: Dear reviewer, thank you very much for your constructive comments and valuable suggestions. Your critical insights have significantly contributed to improving the quality of our manuscript. We have revised the manuscript in accordance with your comments as well as those from Reviewer 4. Please kindly refer to the revised sections corresponding to each comment.

2. Line 67: The authors have revised the phrasing of associations to indicate relationships between variables instead of categories, addressing the previous reviewer’s concern. However, minor rewording for clarity would enhance readability. Dear reviewer, than you in advance for your valuable suggestion. We initially mentioned each category of variables showing statistically significant associations to provide clearer specification and better understanding for readers. However, in response to your constructive feedback and that of Reviewer 4, we have revised the manuscript by replacing the specific categories with the corresponding variables throughout the document, please kindly see on the revised manuscript on page 3, lines 67-71 and pages 17-18, lines 303-309.

3. Line 73-76: The recommendation section mentions targeting "older women, those with formal education, non-working women, individuals who spend time watching television and using the internet, urban residents, female household heads, and women in the wealthiest households."

o This contradicts the previous reviewer's concern that the highest socioeconomic group may not be the most urgent target (previous reviewer's comment). The phrasing should be revised to reflect a more balanced intervention approach. Dear reviewer, thank you for your constructive suggestion. We revised the recommendation as per your valuable guidance, please kindly see on page 4, lines 77-78.

4. Line 92-93: The statement about overweight/obesity increasing among lower economic groups over time should be further clarified. Does this apply to all LMICs or only a subset? Dear reviewer, thank you for your valuable suggestion. This finding is applicable to all 39 low- and middle-income countries included in the study. We have revised the section in accordance with your recommendation, please see on page 5, lines 93-95.

5. Line 131-137: The justification for the study’s objective is clear, but it would help to specify how this study improves on existing literature in terms of methodology or scope. Dear reviewer, thank you in advance for your insightful feedback. We have revised the section in line with your valuable suggestion, please kindly see on page 6, lines 133-138.

6. Line 145-146: The sentence listing countries has been revised, addressing the previous reviewer’s concern about the use of “includes.” Dear reviewer, thank you very much for your confirmation.

7. Line 223-225: The classification of maternal mortality cut-offs using WHO guidelines has been included. This addresses the previous reviewer’s request to explain why African countries dominate the dataset. However, a brief mention of why high maternal mortality is particularly relevant to obesity research would strengthen the rationale. Dear reviewer, thank you in advance for your constructive suggestion, really your valuable comments are significantly improving the quality of our manuscript.

The reason of focusing on those selected countries having high, very high or extremely high maternal mortality was due to the women in such countries seeks an urgent attention to avert causes of morbidity and mortality. We revised the manuscript as per your insightful comment, please kindly see on page 10, lines 232-234.

8. Line 240-241: The authors checked multicollinearity (VIF < 5) to ensure that age and number of living children could be analyzed independently. This adequately addresses the previous reviewer’s concern. However, it may be beneficial to explain how interaction effects were considered, if applicable. Dear reviewer, thank you for your question. We believe that testing the assumption of multicollinearity is sufficient for our analysis, as we did not hypothesize that the combined effect of the independent variables would differ from the sum of their individual effects

9. Line 272-275: The pooled prevalence of overweight/obesity is well reported. However, it might be useful to include a brief mention of which factors contribute to variations between countries to strengthen the analysis. Dear reviewer, thank you for your comment. However, our objective was to determine the magnitude and associated factors of the dependent variable, rather than to explore the reasons behind the variations in prevalence across the studied countries. Additionally, investigating the causes of such variations would require a separate study with a distinct methodological approach.

10. Line 279-282: The discussion of ICC and MOR in the null model is useful. However, the interpretation of ICC (19%) could be expanded to explain whether this level suggests a strong or moderate clustering effect. Dear reviewer, thank you very much for your insightful comment. We revised the manuscript according to your valuable comment, please kindly see on page 17, lines 290-291.

11. Line 296-300: The separation of individual and community-level factors is well done. However, the inclusion of community poverty as a determinant still does not fully address the previous reviewer’s concern about targeting higher-income groups. The discussion should acknowledge this limitation. Dear reviewer, thank you for your concern, however, what we are understanding here that the term “lower community poverty” is a confusing phrase. Due to this reason and as per your final recommendation, we modified this phrase as follows:

1. Low community poverty= high community economic status

2. High community poverty=low community economic status

Therefore, we hope this phrase replacement will overcome the confusion and we revised the entire document with this new adjustment, please kindly see the revised manuscript.

12. Line 373-376: The authors have moved the discussion of internet use as a proxy for higher socioeconomic status to the discussion section instead of the summary of results. This revision is appropriate. The authors could further acknowledge that internet use itself is not a direct cause of obesity, but rather a proxy for wealth and lifestyle behaviours. Dear reviewer, thank you for your critical insight. We have revised the manuscript as per your valuable recommendation, please kindly see on page 25, lines 388-389.

13. Line 386-390: The revision addressing the difference between obesity patterns in developing vs. developed countries is well-written. However, the phrase "low community poverty" is still potentially misleading. Consider replacing it with "higher economic status at the community level." Dear reviewer, thank you very much for your insightful guidance. We truly appreciate your valuable comments, which have significantly improved the quality of our manuscript. We have made the revisions as per your recommendations; please kindly refer to the updated version.

14. Line 406-408: The authors acknowledge the possibility of reverse causation regarding marital status and obesity and provide a brief discussion on this. This improves the credibility of their analysis. Dear reviewer, thank you for your understanding.

15. Statistical Models and Interpretation (Table 4, Multilevel Logistic Regression Analysis, Page 30-31): The authors provided a detailed breakdown of their models, but further discussion on effect sizes and practical implications of findings could enhance clarity. Dear reviewer, thank you very much for your insightful comment. We have discussed the effect sizes and practical implications of the selected variables at the discussion section, please kindly see on pages 23, lines 344-347; page 25, lines 389-391; pages 25-26, lines 400-403; page 26, lines 417-420; and page 27, lines 436-438.

16. Recommendations:

17. • The discussion still suggests that the wealthiest groups should be the focus of intervention, contradicting trends in lower-income groups. Further justification with stronger evidence would help. Dear reviewer, thank you for your recommendation. Since we modified the confusing phrase “low community poverty”, we hope the idea that seems contradiction was solved.

18. The term "low community poverty" is misleading and should be replaced. Dear reviewer, thank you very much for your insightful recommendation. We have replaced it with other appropriate and clear phrase, please kindly see the entire document.

19. ICC interpretation should be more explicit about its strength. Dear reviewer, thank you in advance for your insightful view. We have done it as per your valuable recommendation, please kindly see on page 17, lines 290-291.

20. With these minor refinements, the manuscript would be significantly improved. The methodological rigor and data analysis remain sound, making the study a valuable contribution to public health research in LMICs.

Overall, the authors have made most of the necessary changes, but a final round of refinements would improve readability and interpretation. Dear reviewer, thank you sincerely for your thoughtful comments, suggestions, and critical insights. We truly appreciate your contribution to enhancing the quality of our manuscript. We have made every effort to address your valuable feedback and have worked diligently to improve the manuscript accordingly.

---

## [Decision Letter · Decision Letter 4]

13 Aug 2025

PONE-D-23-42732R4Pooled prevalence and factors of overweight/obesity among women of reproductive age in low and middle-income countries with high maternal mortality:  A multi-level analysis of recent Demographic and Health SurveysPLOS ONE

Dear Dr. Geberu,

Thank you for submitting your manuscript to PLOS ONE. After careful consideration, we feel that it has merit but does not fully meet PLOS ONE’s publication criteria as it currently stands. Therefore, we invite you to submit a revised version of the manuscript that addresses the points raised during the review process.

We look forward to receiving your revised manuscript.

Kind regards,

Zhaoxia Liang

Academic Editor

PLOS ONE

Journal Requirements:

Reviewers' comments:

Reviewer's Responses to Questions

**Comments to the Author**

1. If the authors have adequately addressed your comments raised in a previous round of review and you feel that this manuscript is now acceptable for publication, you may indicate that here to bypass the “Comments to the Author” section, enter your conflict of interest statement in the “Confidential to Editor” section, and submit your "Accept" recommendation.

Reviewer #4: (No Response)

Reviewer #5: All comments have been addressed

2. Is the manuscript technically sound, and do the data support the conclusions?

Reviewer #4: Yes

Reviewer #5: Yes

3. Has the statistical analysis been performed appropriately and rigorously? 

Reviewer #4: Yes

Reviewer #5: Yes

4. Have the authors made all data underlying the findings in their manuscript fully available?

Reviewer #4: Yes

Reviewer #5: Yes

5. Is the manuscript presented in an intelligible fashion and written in standard English?

Reviewer #4: No

Reviewer #5: (No Response)

6. Review Comments to the Author

Reviewer #4: Thank you for repsonding to my previous comments. However, in doing so, the English is now in may places unreadable (eg. lines 67-68; and lines 223-5 make no sense at all). It would help if the English were checked for clarity and grammatical correctness.

Reviewer #5: After a detailed review of the revised manuscript, it is evident that the authors have thoroughly addressed the comments and recommendations raised by the previous reviewers and the editorial team. Key improvements and responses are summarized below:

1. The authors appropriately justified the use of BMI given the constraints of the DHS dataset, while acknowledging its limitations as a proxy for adiposity. This has been clearly articulated in the limitation section.

2. The manuscript now consistently discusses factors at the variable level, rather than focusing on specific categories (e.g., age, marital status, wealth index), addressing a critical concern of Reviewer 4. Revisions are evident in the abstract, results, and discussion

3. Table 4 has been reformatted to reflect valid statistical comparisons at the variable level using a method inspired by Tukey’s approach. Bold formatting and significance stars have been removed in accordance with statistical best practices.

4. The authors revised the interpretation and recommendations to better balance the focus between higher and lower socioeconomic groups, with added clarity on odds ratios and potential intervention targets.

5. The term “low community poverty” has been corrected throughout, and definitions of community-level factors have been clarified. Multi-collinearity checks were conducted; while interaction effects were not modeled, justification was provided based on study scope.

6. Improvements have been made in terminology, clarity, and presentation of results, including a more balanced discussion of causality (e.g., marital status) and the age-obesity relationship.

All major and minor points raised in previous rounds of peer review appear to have been addressed by the authors. The revised manuscript demonstrates significant improvement and is fit for publication. However, a final proofing pass is advisable before publication to correct minor issues that do not affect scientific integrity.

Here is a line-by-line review highlighting minor textual or formatting issues that can be improved before publication. These are not critical but will enhance clarity, consistency, and professionalism of the manuscript.

• Line 75, the phrase “...spend time to watch television and using the internet...” should be rephrased for grammatical consistency. A more appropriate version would be: “...spend time watching television and using the internet...”.

• Line 78, the expression “optimum care for women from the wealthiest households” is vague. It would be clearer to say: “appropriate health interventions for women from the wealthiest households.”

• Line 137, consider refining the phrase “a more nuanced understanding” to “an enhanced understanding” for a slightly more formal tone.

• Line 168, the text mentions “...had given birth in the last five years...” but does not clarify why postpartum women were excluded. Providing this rationale would improve the reader’s comprehension of the study’s scope.

• Lines 223–224, the phrase “Community poverty was categorized as high community economic status...” is confusing. A clearer version would be: “Communities were categorized as having high or low economic status based on the proportion of women in the poorest wealth quintiles.”

• Throughout the manuscript, phrases like “women age”, “women occupation” “women marital status”, and “women educational status” are grammatically incorrect. These should be changed to “women’s age”, “women’s occupation”, “women’s marital status” and “women’s educational status”, respectively.

7. PLOS authors have the option to publish the peer review history of their article (what does this mean? ). If published, this will include your full peer review and any attached files.

**Do you want your identity to be public for this peer review?** For information about this choice, including consent withdrawal, please see our Privacy Policy .

Reviewer #4: No

Reviewer #5: **Yes: ** Dr. Priyanka Airi Goyal

---

## [Author Response · Author response to Decision Letter 5]

25 Aug 2025

Point-by-point response for editor’s and reviewers' comments

Sr.No Editor’s and Reviewers’ comments and suggestions Authors' responses

Editor’s comments and suggestions

1. Thank you for submitting your manuscript to PLOS ONE. After careful consideration, we feel that it has merit but does not fully meet PLOS ONE’s publication criteria as it currently stands. Therefore, we invite you to submit a revised version of the manuscript that addresses the points raised during the review process. Dear Editor, Thank you for your valuable support in enhancing the quality of our manuscript. We have carefully reviewed the reviewers’ comments and have devoted substantial time and research effort to address each point comprehensively. The entire manuscript has been revised accordingly.

2. Journal Requirements:

Dear Editor, Thank you in advance for your guidance. We have carefully reviewed all the journal requirements and thoroughly addressed your suggestions. Regarding the citation of retracted articles, we have verified each cited article using the Retraction Watch database (https://retractiondatabase.org/RetractionSearch.aspx?) to ensure that no retracted journals have been cited, therefore, we didn’t cite retracted papers. During the review process, we got some miner editorial problem on the reference section and we revised it accordingly. Thank you in advance for your insightful feedback!

Reviewer four’s comments and suggestions

1. 5. Is the manuscript presented in an intelligible fashion and written in standard English?

Reviewer #4: No

Dear reviewer, Thank you for your comments regarding the language of our manuscript. We have invited a language expert and thoroughly revised the entire manuscript to improve its clarity and readability. Kindly refer to the updated version for your review.

2. Thank you for repsonding to my previous comments. However, in doing so, the English is now in may places unreadable (eg. lines 67-68; and lines 223-5 make no sense at all). It would help if the English were checked for clarity and grammatical correctness. Dear reviewer, Thank you again for your comments regarding the language of our manuscript. We have invited a language expert and thoroughly revised your specific area comments and entire manuscript to improve its clarity and readability. Please kindly see on page 3, lines 67-68, on page 11 lines 229-231 and refer to the entire updated version.

Reviewer five’s comments and suggestions

1. After a detailed review of the revised manuscript, it is evident that the authors have thoroughly addressed the comments and recommendations raised by the previous reviewers and the editorial team. Key improvements and responses are summarized below:

1. The authors appropriately justified the use of BMI given the constraints of the DHS dataset, while acknowledging its limitations as a proxy for adiposity. This has been clearly articulated in the limitation section.

2. The manuscript now consistently discusses factors at the variable level, rather than focusing on specific categories (e.g., age, marital status, wealth index), addressing a critical concern of Reviewer 4. Revisions are evident in the abstract, results, and discussion

3. Table 4 has been reformatted to reflect valid statistical comparisons at the variable level using a method inspired by Tukey’s approach. Bold formatting and significance stars have been removed in accordance with statistical best practices.

4. The authors revised the interpretation and recommendations to better balance the focus between higher and lower socioeconomic groups, with added clarity on odds ratios and potential intervention targets.

5. The term “low community poverty” has been corrected throughout, and definitions of community-level factors have been clarified. Multi-collinearity checks were conducted; while interaction effects were not modeled, justification was provided based on study scope.

6. Improvements have been made in terminology, clarity, and presentation of results, including a more balanced discussion of causality (e.g., marital status) and the age-obesity relationship.

All major and minor points raised in previous rounds of peer review appear to have been addressed by the authors. The revised manuscript demonstrates significant improvement and is fit for publication. However, a final proofing pass is advisable before publication to correct minor issues that do not affect scientific integrity. Dear reviewer, Thank you in advance for confirming that we have addressed your previous comments and concerns. Your insightful feedback has greatly contributed to improving the quality of our manuscript. We have also conducted a final language revision with the assistance of a language expert and completed a thorough proofreading. Please kindly refer to the updated manuscript.

2. Here is a line-by-line review highlighting minor textual or formatting issues that can be improved before publication. These are not critical but will enhance clarity, consistency, and professionalism of the manuscript.

• Line 75, the phrase “...spend time to watch television and using the internet...” should be rephrased for grammatical consistency. A more appropriate version would be: “...spend time watching television and using the internet...”. Dear reviewer, Thank you very much for your valuable suggestion. We have revised the manuscript accordingly, please kindly see on page 4, line 76.

3. Line 78, the expression “optimum care for women from the wealthiest households” is vague. It would be clearer to say: “appropriate health interventions for women from the wealthiest households.” Dear reviewer, Thank you very much for your valuable suggestion. We have revised the manuscript accordingly, please kindly see on page 4, lines 78-79.

4. Line 137, consider refining the phrase “a more nuanced understanding” to “an enhanced understanding” for a slightly more formal tone. Dear reviewer, Thank you for your important suggestion. We have revised the manuscript accordingly, please kindly see on page 6, line 138.

5. Line 168, the text mentions “...had given birth in the last five years...” but does not clarify why postpartum women were excluded. Providing this rationale would improve the reader’s comprehension of the study’s scope. Dear reviewer, thank you in advance for your insightful feedback. We have clearly addressed and revised the section in line with your valuable suggestion, please kindly see on page 8, lines 170-174.

6. Lines 223–224, the phrase “Community poverty was categorized as high community economic status...” is confusing. A clearer version would be: “Communities were categorized as having high or low economic status based on the proportion of women in the poorest wealth quintiles.” Dear reviewer, thank you very much for your suggestion. We considered your invaluable comment to render clear meaning and to avoid confusion, please kindly see on page 11, lines 229-230.

7. Throughout the manuscript, phrases like “women age”, “women occupation” “women marital status”, and “women educational status” are grammatically incorrect. These should be changed to “women’s age”, “women’s occupation”, “women’s marital status” and “women’s educational status”, respectively. Dear reviewer, thank you in advance for your constructive suggestion, really your valuable comments are significantly improving the quality of our manuscript. We revised the entire section according to your valuable suggestion, please kindly see the updated manuscript.

---

## [Editor Report · Decision Letter 5]

18 Sep 2025

Pooled prevalence and factors of overweight/obesity among women of reproductive age in low and middle-income countries with high maternal mortality:  A multi-level analysis of recent Demographic and Health Surveys

PONE-D-23-42732R5

Dear Dr. Geberu,

We’re pleased to inform you that your manuscript has been judged scientifically suitable for publication and will be formally accepted for publication once it meets all outstanding technical requirements.

Kind regards,

Zhaoxia Liang

Academic Editor

PLOS ONE
---

## [Editor Report · Acceptance letter]

PONE-D-23-42732R5

PLOS ONE

Dear Dr. Geberu,

I'm pleased to inform you that your manuscript has been deemed suitable for publication in PLOS ONE. Congratulations! Your manuscript is now being handed over to our production team.

Kind regards,

on behalf of

Dr. Zhaoxia Liang

Academic Editor

PLOS ONE